# A Variational Perspective on Solving Inverse Problems with Diffusion Models

**Morteza Mardani, Jiaming Song, Jan Kautz, Arash Vahdat**
NVIDIA Inc.
`mmardani,jiamings,jkautz,avahdat@nvidia.com`

## Abstract

Diffusion models have emerged as a key pillar of foundation models in visual domains. One of their critical applications is to universally solve different downstream inverse tasks via a single diffusion prior without re-training for each task. Most inverse tasks can be formulated as inferring a posterior distribution over data (e.g., a full image) given a measurement (e.g., a masked image). This is however challenging in diffusion models since the nonlinear and iterative nature of the diffusion process renders the posterior intractable. To cope with this challenge, we propose a variational approach that by design seeks to approximate the true posterior distribution. We show that our approach naturally leads to regularization by denoising diffusion process (RED-diff) where denoisers at different timesteps concurrently impose different structural constraints over the image. To gauge the contribution of denoisers from different timesteps, we propose a weighting mechanism based on signal-to-noise-ratio (SNR). Our approach provides a new variational perspective for solving inverse problems with diffusion models, allowing us to formulate sampling as stochastic optimization, where one can simply apply off-the-shelf solvers with lightweight iterates. Our experiments for various linear and nonlinear image restoration tasks demonstrate the strengths of our method compared with state-of-the-art sampling-based diffusion models. The code is available online [1].

## 1 Introduction

Diffusion models such as Stable diffusion (Rombach et al., 2021) are becoming an integral part of nowadays visual foundation models. An important utility of such diffusion models is to use them as prior distribution for sampling in various downstream inverse problems appearing for instance in image restoration and rendering. This however demands samplers that are (i) *universal* and adaptive to various tasks without re-training for each individual task, and (ii) efficient and easy to tune.

There has been a few recent attempts to develop universal samplers for inverse problems; (Kawar et al., 2022a; Song et al., 2023; Chung et al., 2022a; Kadkhodaie & Simoncelli, 2021; Graikos et al., 2022) to name a few. DDRM (Kawar et al., 2022a) was initially introduced to extend DDPM (Ho et al., 2020) to handle linear inverse problems. It relies on SVD to integrate linear observations into the denoising process. DDRM however needs many measurements to work. Later on, ΠGDM was introduced (Song et al., 2023) to enhance DDRM. The crux of ΠGDM is to augment the denoising diffusion score with the guidance from linear observations through inversion. In a similar vein, DPS (Chung et al., 2022a) extends the score modification framework to general (nonlinear) inverse problems. The score modification methods in DPS and ΠGDM, however, heavily resort to approximations. In essence, the nonlinear and recursive nature of the backward diffusion process renders the posterior distribution quite intractable and multimodal. However, DPS and ΠGDM rely on a simple unimodal approximation of the score which is a quite loose approximation at many steps of the diffusion process.

To sidestep the challenges for posterior score approximation, we put forth a fundamentally different approach based on variational inference (Blei et al., 2017; Ahmed et al., 2012; Hoffman et al., 2013). Adopting the denoising diffusion model as our data prior and representing the measurement model as a likelihood, we use variational inference to infer the posterior distribution of data given the observations. Our method essentially matches modes of data distribution with a Gaussian distribution

---

[1] `https://github.com/NVlabs/RED-diff`

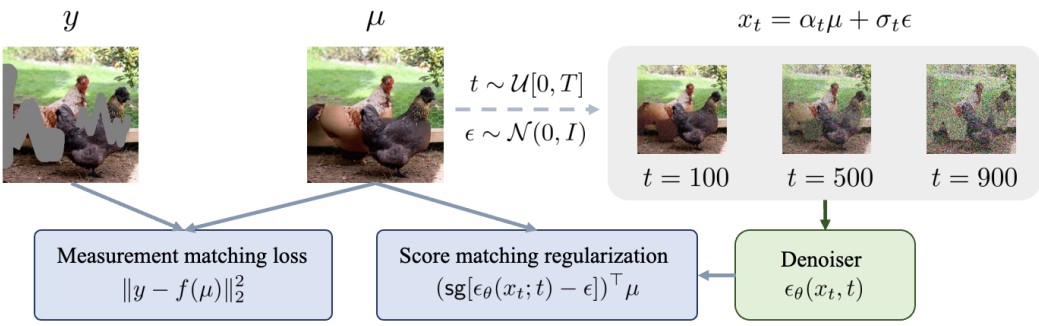

Figure 1: The schematic diagram of our proposed variational sampler (RED-diff). The forward denoising diffusion process gradually adds noise to the estimate $\mu$. The denoisers of the backward diffusion process apply score-matching regularization to the measurement matching loss. The refined estimate using optimization is then fed back to the forward process and the process repeats.

using KL divergence. That leads to a simple (weighted) score-matching criterion that regularizes the measurement matching loss from observations via denoising diffusion process. Interestingly, the score-matching regularization admits an interpretable form with simple gradients; see Fig. 1.

This resembles the regularization-by-denoising (RED) framework by Romano et al. (2016), where denoisers at different stages of the diffusion process impose different structural constraints from high-level semantics to fine details. This is an important connection that views sampling as stochastic optimization. As a result, one can simply deploy the rich library of off-the-shelf optimizers for sampling which makes inference efficient, interpretable, and easy to tweak. We coin the term RED-diff to name our method. It is however worth noting that our framework differs from RED in several aspects: $(i)$ we derive our objective from a principled variational perspective that is well studied and understood, and $(ii)$ our regularization uses feedback from all the diffusion steps with different noise levels while RED uses a single denoising model.

For the success of the score matching regularization, denoisers at different timesteps need to be weighted properly. To do so, we propose a weighting mechanism based on densoing SNR at each timestep that upweights the earlier steps in the reverse diffusion process and down-weights the later timesteps. To verify the proposed idea, we conduct experiments and ablations for various linear and nonlinear inverse problems. Our main insights indicate that: $(i)$ RED-diff achieves superior image fidelity and perceptual quality compared with state-of-the-art samplers for image inverse problems; $(ii)$ RED-diff has lightweight iterates with no score Jacobian involved as in DPS and $\Pi$GDM, and as a result, it is more memory efficient and GPU friendly; $(iii)$ Our ablation studies suggest that the optimizer parameters such as learning rate and the number of steps are suitable knobs to tweak the trade-off between fidelity and perceptual quality.

**Contributions**. All in all, the main contributions of this paper are summarized as follows:

- We propose, RED-diff, a variational approach for general inverse problems, by introducing a rigorous maximum-likelihood framework that mitigates the posterior score approximation involved in recent $\Pi$GDM (Song et al., 2023) and DPS (Chung et al., 2022a)
- We establish a connection with regularization-by-denoising (RED) framework (Romano et al., 2016), which allows to treat sampling as stochastic optimization, and thus enables off-the-shelf optimizers for fast and tunable sampling
- We propose a weighting mechanism based on denoising SNR for the diffusion regularization
- We conduct extensive experiments for various linear and nonlinear inverse problems that show superior quality and GPU efficiency of RED-diff against state-of-the-art samplers such as $\Pi$GDM and DPS. Our ablations also suggest key insights about tweaking sampling and optimization to generate good samples.

## 2 RELATED WORKS

Our work is primarily related to the following lines of work in the context of diffusion models.

**Diffusion models for inverse problems**: There are several recent works to apply diffusion models in a plug-and-play fashion to inverse problems in various domains such as natural images (Kadkhodaie

& Simoncelli, 2021; Jalal et al., 2021; Kawar et al., 2022a; Song et al., 2023; Chung et al., 2022a;b; Graikos et al., 2022; Chung et al., 2023a;b), medical images (Jalal et al., 2021), and audio processing (Kong et al., 2020). We primarily focus on images, where these works primarily differ in the way that they handle measurements. As some of the early works, Kadkhodaie & Simoncelli (2021) and Jalal et al. (2021) adopt Langevine dynamics for linear inverse problems and integrate the observation guidance via either projection (Kadkhodaie & Simoncelli, 2021), or gradient of the least-squares fidelity (Jalal et al., 2021). Some other works adopt DDPM (Ho et al., 2020) diffusion and alternate between diffusion denoising and projection steps (Choi et al., 2021; Chung et al., 2022c). The iterations however can accumulate error that pushes the trajectory off the prior manifold, and thus MCG method (Chung et al., 2022b) proposes an additional correction term inspired by the manifold constraint to keep the iterations close to the manifold. DDRM (Kawar et al., 2022a) extends DDPM to solve linear inverse problems using matrix SVD, but it fails for limited measurements.

To address this shortcoming, recent methods aim to provide guidance by differentiating through the diffusion model in the form of reconstruction guidance (Ho et al., 2022), which is further extended in DPS (Chung et al., 2022a) to nonlinear inverse problems. ΠGDM (Song et al., 2023) introduces pseudoinverse guidance that improves the guidance approximation by inverting the measurement model. Its scope is however limited to linear and certain semi-linear tasks (such as JPEG (Kawar et al., 2022b)). However, both ΠGDM and DPS heavily rely on an approximation of the intractable posterior score, which is quite crude for non-small noise levels at many steps of the diffusion process. Note also that, a different method has also been recently proposed by Graikos et al. (2022), which regularizes the reconstruction term of inverse problems with the diffusion error loss. This is similar to the traditional plug-and-play prior ($P^3$) approach for inverse problems (Venkatakrishnan et al., 2013) that roots back to ADMM optimization (Boyd et al., 2011). Our method is however closer in spirit to the RED framework, which offers more flexibility for optimizer and tuning; see e.g., (Romano et al., 2016; Cohen et al., 2021).

**Diffusion models for 3D**: A few recent works have adopted distillation loss optimization to generate 3D data from 2D diffusion priors, which is related to our view of treating sampling as optimization. For instance, DreamFusion (Poole et al., 2022) and ProfilicDreamer (Wang et al., 2023) adopt a probability density distillation loss as the criterion for text-to-3D generation. Followup works include SparseFusion (Zhou & Tulsiani, 2022) that generates 3D given a few (e.g. just two) segmented input images with known relative pose, and NeuralLift-360 (Xu et al., 2022) that lifts a single 2D image to 3D. All these methods use a distillation loss, that bears resemblance with our (unweighted) denoising regularization. However, they aim to optimize for a parametric 3D NeRF model that is fundamentally different from our goal.

## 3 BACKGROUND

In this section, we first review diffusion models in Section 3.1 and we discuss how they are used for solving inverse problems in Section 3.2.

### 3.1 DENOISING DIFFUSION MODELS

Diffusion models (Sohl-Dickstein et al., 2015; Ho et al., 2020; Song et al., 2021b) consist of two processes: a forward process that gradually adds noise to input images and a reverse process that learns to generate images by iterative denoising. Formally the forward process can be expressed by the variance preserving stochastic differential equation (VP-SDE) (Song et al., 2021b) $dx = -\frac{1}{2}\beta(t)xdt + \sqrt{\beta(t)}dw$ for $t \in [0, T]$ where $\beta(t) := \beta_{\min} + (\beta_{\max} - \beta_{\min})\frac{t}{T}$ rescales the time variable, and $dw$ is the standard Wiener process. The forward process is designed such that the distribution of $x_T$ at the end of the process converges to a standard Gaussian distribution (i.e., $x_T \sim \mathcal{N}(0, I)$). The reverse process is defined by $dx = -\frac{1}{2}\beta(t)xdt - \beta(t)\nabla_{x_t}\log p(x_t) + \sqrt{\beta(t)}d\bar{w}$ where $\nabla_{x_t}\log p(x_t)$ is *the score function* of diffused data at time $t$, and $d\bar{w}$ is the reverse standard Wiener process.

Solving the reverse generative process requires estimating the score function. In practice, this is done by sampling from the forward diffusion process and training the score function using the denoising score-matching objective (Vincent, 2011). Specifically, diffused samples are generated by:

$$x_t = \alpha_t x_0 + \sigma_t \epsilon, \quad \epsilon \sim \mathcal{N}(0, I), \quad t \in [0, T] \tag{1}$$

where $x_0 \sim p_{\text{data}}$ is drawn from data distribution, $\sigma_t = 1 - e^{-\int_0^t \beta(s)ds}$, and $\alpha_t = \sqrt{1 - \sigma_t^2}$. Let's denote the parameterized score function (i.e., diffusion model) by $\epsilon_\theta(x_t; t) \approx -\sigma_t \nabla_{x_t}\log p(x_t)$ with

parameters $\theta$, we can train $\epsilon_\theta(x_t; t)$ with a mixture of Euclidean losses, such as

$$\min_\theta \mathbb{E}_{x_0 \sim p_{\text{data}}(x_0), \epsilon \sim \mathcal{N}(0, I), t \sim \mathcal{U}[0, T]} \left[ ||\epsilon - \epsilon_\theta(x_t; t)||_2^2 \right].$$

Other loss-weighting functions for $t$ can be used as well. Given a trained score function, samples can be generated using DDPM (Ho et al., 2020), DDIM (Song et al., 2020), or other solvers (Lu et al., 2022; Zhang & Chen, 2022; Dockhorn et al., 2022).

## 3.2 SCORE APPROXIMATION FOR INVERSE PROBLEMS

Inverse problems can be formulated as finding $x_0$ from a (nonlinear and noisy) observation:

$$y = f(x_0) + v, \quad v \sim \mathcal{N}(0, \sigma_v^2 I) \tag{2}$$

where the forward (a.k.a measurement) model $f$ is known. In many applications, such as inpainting, this is a severely ill-posed task that requires a strong prior to find a plausible solution. Our goal is to leverage the prior offered by (pretrained) diffusion models, in a plug-and-play fashion, to efficiently sample from the conditional posterior. Let's denote the prior distributions imposed by diffusion models as $p(x_0)$. The measurement models can be represented by $p(y|x_0) := \mathcal{N}(f(x_0), \sigma_v^2)$. The goal of solving inverse problems is to sample from the posterior distribution $p(x_0|y)$.

As we discussed in the previous section, diffusion models rely on the estimated score function to generate samples. In the presence of the measurements $y$, they can be used for generating plausible $x_0 \sim p(x_0|y)$ as long as an approximation of the conditional score for $p(x_t|y)$ over all diffusion steps is available. This is the idea behind $\Pi$GDM (Song et al., 2023) and DPS (Chung et al., 2022a). Specifically, the conditional score for $p(x_t|y)$ based on Bayes rule is simply obtained as

$$\nabla_x \log p(x_t|y) = \nabla_x \log p(y|x_t) + \nabla_x \log p(x_t) \tag{3}$$

The overall score is a superposition of the model likelihood and the prior score. While $\nabla_x \log p(x_t)$ is easily obtained from a pretrained diffusion model, the likelihood score is quite challenging and intractable to estimate without any task-specific training. This can be seen from the fact that $p(y|x_t) = \int p(y|x_0)p(x_0|x_t)dx_0$. Although $p(y|x_0)$ takes a simple Gaussian form, the denoising distribution $p(x_0|x_t)$ can be highly complex and multimodal (Xiao et al., 2022). As a result, $p(y|x_t)$ can be also highly complex. To sidestep this, prior works (Song et al., 2023; Chung et al., 2022a; Kadkhodaie & Simoncelli, 2021; Ho et al., 2022) resort to Gaussian approximation of $p(x_0|x_t)$ around the MMSE estimate

$$\mathbb{E}[x_0|x_t] = \frac{1}{\alpha_t}(x_t - \sigma_t \epsilon_\theta(x_t, t)). \tag{4}$$

## 4 VARIATIONAL DIFFUSION SAMPLING

In this section, we introduce our variational perspective on solving inverse problems. To cope with the shortcomings of previous methods for sampling the conditional posterior $p(x_0|y)$, we propose a variational approach based on KL minimization

$$\min_q KL\big(q(x_0|y)||p(x_0|y)\big) \tag{5}$$

where $q := \mathcal{N}(\mu, \sigma^2 I)$ is a variational distribution. The distribution $q$ seeks the dominant mode in the data distribution that matches the observations. It is easy to show that the KL objective in Eq. 5 can be expanded as

$$KL\big(q(x_0|y)\|p(x_0|y)\big) = \underbrace{-\mathbb{E}_{q(x_0|y)}\big[\log p(y|x_0)\big] + KL\big(q(x_0|y)\|p(x_0)\big)}_{\text{term (i)}} + \underbrace{\log p(y)}_{\text{term (ii)}} \tag{6}$$

where term (i) is the variational bound that is often used for training variational autoencoders (Kingma & Welling, 2013; Rezende et al., 2014) and term (ii) is the observation likelihood that is *constant* w.r.t. $q$. Thus, to minimize the KL divergence shown in Eq. 5 w.r.t. $q$, it suffices to minimize the variational bound (term (i)) in Eq. 6 w.r.t. $q$. This brings us to the next claim.

**Proposition 1**. *Assume that the score is learned exactly, i.e., $\epsilon_\theta(x_t; t) = -\sigma_t \nabla_{x_t} \log p(x_t)$. Then, the KL minimization w.r.t $q$ in Eq. 5 is equivalent to minimizing the variational bound (term (i) in Eq. 6), that itself obeys the score matching loss:*

$$\min_{\{\mu, \sigma\}} \mathbb{E}_{q(x_0|y)} \left[ \frac{\|y - f(x_0)\|_2^2}{2\sigma_v^2} \right] + \int_0^T \tilde{\omega}(t) \mathbb{E}_{q(x_t|y)} \left[ \left\| \nabla_{x_t} \log q(x_t|y) - \nabla_{x_t} \log p(x_t) \right\|_2^2 \right] dt, \tag{7}$$

*where $q(x_t|y) = \mathcal{N}(\alpha_t\mu, (\alpha_t^2\sigma^2 + \sigma_t^2)I)$ produces samples $x_t$ by drawing $x_0$ from $q(x_0|y)$ and applying the forward process in Eq. 1, and $\tilde{\omega}(t) = \beta(t)/2$ is a loss-weighting term.*

Above, the first term is the measurement matching loss (i.e., reconstruction loss) obtained by the definition of $p(y|x_0)$, while the second term is obtained by expanding the KL term in terms of the score-matching objective as shown in (Vahdat et al., 2021; Song et al., 2021a), and $\tilde{\omega}(t) = \beta(t)/2$ is a weighting based on maximum likelihood (the proof is provided in the supplementary material). The second term can be considered as a score-matching regularization term imposed by the diffusion prior. The integral is evaluated on a diffused trajectory, namely $x_t \sim q(x_t|y)$ for $t \in [0, T]$, which is the forward diffusion process applied to $q(x_0|y)$. Since $q(x_0|y)$ admits a simple Gaussian form, we can show that $q(x_t|y)$ is also a Gaussian in the form $q(x_t|y) = \mathcal{N}(\alpha_t\mu, (\alpha_t^2\sigma^2 + \sigma_t^2)I)$ (see (Vahdat et al., 2021)). Thus, the score function $\nabla_{x_t}\log q(x_t|y)$ can be computed analytically.

Assuming that the variance of the variational distribution is a small constant value near zero (i.e., $\sigma \approx 0$), the optimization problem in Eq. 7 can be further simplified to:

$$\min_{\mu} \underbrace{\|y - f(\mu)\|^2}_{\text{recon}} + \underbrace{\mathbb{E}_{t,\epsilon}\big[2\omega(t)(\sigma_v/\sigma_t)^2||\epsilon_\theta(x_t; t) - \epsilon||_2^2\big]}_{\text{reg}}, \tag{8}$$

where $x_t = \alpha_t\mu + \sigma_t\epsilon$. In a nutshell, solving the optimization problem above will find an image $\mu$ that reconstructs the observation $y$ given the measurement model $f$, while having a high likelihood under the prior as imposed by the regularization term.

**Remark [Noiseless observations].** If the observation noise $\sigma_v = 0$, then from equation 6 the reconstruction term boils down to a hard constraint which can be represented as an indicator function $\mathbb{1}_{\{y=f(\mu)\}}$ that is zero when $y = f(\mu)$ and infinity elsewhere. In practice, however we can still use equation 7 with a small $\sigma_v$ as an approximation.

## 4.1 SAMPLING AS STOCHASTIC OPTIMIZATION

The regularized score matching objective Eq. 8 allows us to formulate sampling as optimization for inverse problems. In essence, the ensemble loss over different diffusion steps advocates for stochastic optimization as a suitable sampling strategy.

However, in practice the choice of weighting term $\tilde{\omega}(t)$ plays a key role in the success of this optimization problem. Several prior works on training diffusion models (Ho et al., 2020; Vahdat et al., 2021; Karras et al., 2022; Choi et al., 2022) have found that reweighting the objective over $t$ plays a key role in trading content vs. detail at different diffusion steps which we also observe in our case (more information in Section 4.3). Additionally, the second term in Eq. 8 marked by "reg" requires backpropagating through pretrained score function which can make the optimization slow and unstable. Next, we consider a generic weighting mechanism $\tilde{\omega}(t) = \beta(t)\omega(t)/2$ for a positive-valued function $\omega(t)$, and we show that if the weighting is selected such that $\omega(0) = 0$, the gradient of the regularization term can be computed efficiently without backpropagating through the pretrained score function.

**Proposition 2.** *If $\omega(0) = 0$ and $\sigma = 0$, the gradient of the score matching regularization admits*

$$\nabla_\mu\text{reg}(\mu) = \mathbb{E}_{t\sim\mathcal{U}[0,T],\epsilon\sim\mathcal{N}(0,I)}\big[\lambda_t(\epsilon_\theta(x_t; t) - \epsilon)\big]$$

*where $\lambda_t := \frac{2T\sigma_v^2\alpha_t}{\sigma_t}\frac{d\omega(t)}{dt}$.*

**First-order stochastic optimizers**. Based on the simple expression for the gradient of score-matching regularization in Proposition 2, we can treat time as a uniform random variable. Thus by sampling randomly over time and noise, we can easily obtain unbiased estimates of the gradients. Accordingly, first-order stochastic optimization methods can be applied to search for $\mu$. We list the iterates under Algorithm 1. Note that we define the loss per timestep based on the instantaneous gradient, which can be treated as a gradient of a linear loss. We introduce the notation (sg) as stopped-gradient to emphasize that score is not differentiated during the optimization. The ablations in Section D.4 show that (descending) time stepping from $t = T$ to $t = 0$, as in standard backward diffusion samplers such as DDPM and DDIM, performs better than random time sampling in practice.

**Remark [Non-zero dispersion]**. Note that Proposition 2 derives the gradient for no dispersion case (i.e., $\sigma = 0$) for simplicity. The extension to nonzero dispersion is deferred to the appendix (A.3).

---

**Algorithm 1** Variational sampler (RED-diff)

---

**Input:** $y, f(\cdot), \sigma_v, L, \{\alpha_t, \sigma_t, \lambda_t\}_{t=1}^T$
**Initialize:** $\mu_0$
    **for** $\ell = 1, \ldots, L$ **do**
        $t \sim \mathcal{U}[0, T]$
        $\epsilon \sim \mathcal{N}(0, I_n)$
        $x_t = \alpha_t \mu + \sigma_t \epsilon$
        $loss = \|y - f(\mu)\|^2 + \lambda_t (\mathsf{sg}[\epsilon_\theta(x_t; t) - \epsilon])^\top \mu$
        $\mu \leftarrow \mathsf{OptimizerStep}(loss)$
    **end for**
**Return:** $\mu$

---

## 4.2 REGULARIZATION BY DENOISING

Note that our variational sampler strikes resemblance with the regularization by denoising (RED) framework (Romano et al., 2016). In essence, RED is a flexible way to harness a given denoising engine for treating general inverse problems. RED regularization effectively promotes smoothness for the image according to some image-adaptive Laplacian prior. To better understand the connection with RED, let us look at the loss per timestep of our proposed variational sampler. From the gradient expression in Proposition 2, it is useful to form the loss at timestep $t$ as

$$\|y - f(\mu)\|^2 + \lambda_t (\mathsf{sg}[\epsilon_\theta(x_t; t) - \epsilon])^\top \mu \tag{9}$$

This regularization term resembles RED. A small regularization implies that either the diffusion reaches a fixed point, namely $\epsilon_\theta(x_t; t) = \epsilon$, or the residual only contains noise with no contribution left from the image. It should however be noted that there is no need for Jacobian symmetry, or the assumptions needed in the original RED Romano et al. (2016) since gradient of 9 is naturally $\epsilon_\theta(x_t; t) - \epsilon$ (note the stopped gradient operation $\mathsf{sg}$). Having said that, there are fundamental differences with RED including the generative nature of diffusion prior, and the fact that we use the entire diffusion trajectory for regularization . Nonetheless, we believe this is an important connection to leverage RED utilities for improved sampling of diffusion models in inverse problems. It is also worth commenting that the earlier work by Reehorst & Schniter (2018) also draws connections between RED and score matching based on a single (deterministic) denoiser.

## 4.3 WEIGHTING MECHANISM

In principle, timestep weighting plays a key role in training diffusion models. Different timesteps are responsible for generating different structures ranging from large-scale content in the last timesteps to fine-scale details in the earlier timesteps (Choi et al., 2022). For effective regularization, it is thus critical to properly tune the denoiser weights $\{\lambda_t\}$ in our Algorithm 1. We observed that the regularization term in Eq. 9 is sensitive to noise schedule. For example, in the variance-preserving scenario, it drastically blows up as $t$ approaches zero.

To mitigate the regularization sensitivity to weights, it is more desirable to define the regularization in the signal domain, that is compatible with the fitting term as

$$\|y - f(\mu)\|^2 + \lambda (\mathsf{sg}[\mu - \hat{\mu}_t])^\top \mu, \tag{10}$$

where $\lambda$ is a hyperparameter that balances between the prior and likelihood and $\hat{\mu}_t$ is the MMSE predictor of clean data. Here, we want the constant $\lambda$ to control the trade-off between bias (fit to observations) and variance (fit to prior). In order to come up with the interpretable loss in equation 10, one needs to rescale the noise residual term $\epsilon_\theta(x_t; t) - \epsilon$.

Recall that the denoiser at time $t$ observes $x_t = \alpha_t x_0 + \sigma_t \epsilon$. MMSE estimator also provides denoising as

$$\hat{\mu}_t = \mathbb{E}[\mu | x_t] = \frac{1}{\alpha_t}(x_t - \sigma_t \epsilon_\theta(x_t; t)). \tag{11}$$

Thus, one can show that

$$\mu - \hat{\mu}_t = (\sigma_t / \alpha_t)(\epsilon_\theta(x_t; t) - \epsilon)$$

where we define $\mathrm{SNR}_t := \alpha_t / \sigma_t$ as the signal-to-noise ratio. Accordingly, by choosing $\lambda_t = \lambda / \mathrm{SNR}_t$, we can simply convert the noise prediction formulation in equation 9 to clean data formulation in equation 10.

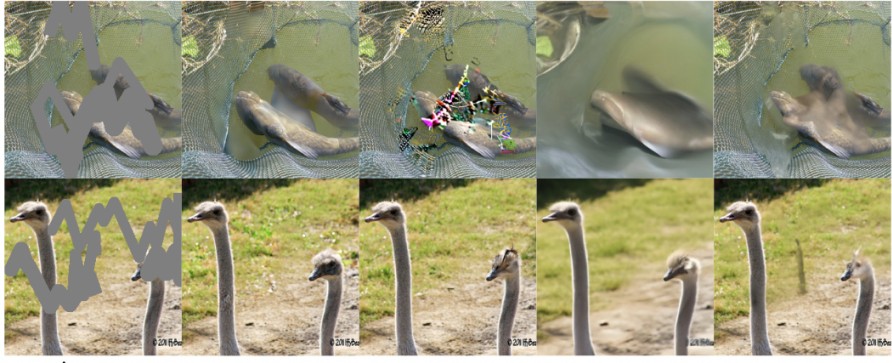

|  input | RED-Diff | ΠGDM | DPS | DDRM |

Figure 2: Comparison of the proposed variational sampler with alternatives for inpainting representative ImageNet examples. Each sampler is tuned for the best performance.

## 5 EXPERIMENTS

In this section, we compare our proposed variational approach, so termed RED-diff, against recent state-of-the-art techniques for solving inverse problems on different image restoration tasks. For the prior, we adopt publicly available checkpoints from the guided diffusion model[2] that is pretrained based on $256 \times 256$ ImageNet (Russakovsky et al., 2015); see details in the appendix. We consider the unconditional version. For the proof of concept, we report findings for various linear and nonlinear image restoration tasks for a 1k subset of ImageNet (Russakovsky et al., 2015) validation dataset[3]. Due to space limitation, we defer more elaborate experiments and ablations to the appendix. Next, we aim to address the following important questions:

- How does the proposed variational sampler (RED-diff) compare with state-of-the-art methods such as DPS, ΠGDM, and DDRM in terms of quality and speed?
- What is a proper sampling strategy and weight-tuning mechanism for the variational sampler?

**Sampling setup**. We adopt linear schedule for $\beta_t$ from $0.0001$ to $0.02$ for $1,000$ timesteps. For simplicity, we always use uniform spacing when we iterate the timestep. For our variational sampler we adopt Adam optimizer with $1,000$ steps, and set the momentum pair $(0.9, 0.99)$ and initial learning rate $0.1$. No weight decay regularization is used. The optimizer is initialized with the degraded image input. We also choose descending time stepping from $t = T$ to $t = 1$ as demonstrated by the ablations later in Section 5.3.2. Across all methods, we also use a batch size of 10 using RTX 6000 Ada GPU with 48GB RAM.

**Comparison**. For comparison we choose state-of-the-art techniques including DPS (Chung et al., 2022a), ΠGDM (Song et al., 2023), and DDRM (Kawar et al., 2022a) as existing alternatives for sampling inverse problems. We tune the hyper-parameters as follows:

- DPS (Chung et al., 2022a): $1,000$ diffusion steps, tuned $\eta = 0.5$ for the best performance;
- ΠGDM (Song et al., 2023): $100$ diffusion steps, tuned $\eta = 1.0$, and observed that ΠGDM does perform worse for 1000 steps;
- DDRM (Kawar et al., 2022a): tested for both 20 and 100 steps, and set $\eta = 0.85$, $\eta_b = 1.0$. DDRM is originally optimized for 20 steps.

For evaluation, we report metrics including Kernel Inception Distance (KID, (Bińkowski et al., 2018)), LPIPS, SSIM, PSNR, and top-1 classifier accuracy of a pre-trained ResNet50 model (He et al., 2015).

### 5.1 IMAGE INPAINTING

For inpainting evaluation, we adopt 1k samples from the ImageNet dataset and random masks from Palette (Saharia et al., 2022). We tune $\lambda = 0.25$ for the SNR-based denoiser weight tuning discussed in Section 4.3. A few representative examples are shown in Fig. 2. For a fair comparison, we choose a relatively hard example in the first row, and an easier one in the bottom row. It is evident that RED-diff identifies the context, and adds the missing content with fine details. ΠGDM however

---

[2]https://github.com/openai/guided-diffusion
[3]https://bit.ly/eval-pix2pix

Table 1: Performance of different samplers for ImageNet inpainting using pretrained unconditional guided diffusion model. For RED-diff we set $lr = 0.5$. For time per step (for each sample) we use the maximum batch size that fits the GPU memory. All methods run on a single NVIDIA RTX 6000 Ada GPU with 48GB RAM.

| Sampler | PSNR(dB) ↑ | SSIM ↑ | KID ↓ | LPIPS ↓ | top-1 ↑ | time per step (sec) ↓ | max batch size ↑ |
|---------|-----------|--------|-------|---------|---------|----------------------|------------------|
| DPS | 21.27 | 0.67 | 15.28 | 0.26 | 58.2 | 0.23 | 15 |
| ΠGDM | 20.30 | 0.82 | 4.50 | 0.12 | 67.8 | 0.24 | 15 |
| DDRM | 20.72 | 0.83 | 2.5 | 0.14 | 68.6 | 0.1 | 25 |
| RED-diff | **23.29** | **0.87** | **0.86** | **0.1** | **72.0** | **0.05** | **30** |

Table 2: Performance of different samplers for nonlinear tasks based on ImageNet data.

| Task | | Phase Retrieval | | HDR | | Deblurring | |
|------|---|-----------------|---|-----|---|------------|---|
| Metrics | | DPS | RED-diff | DPS | RED-diff | DPS | RED-diff |
| PSNR(dB) ↑ | | 9.99 | **10.53** | 7.94 | **25.23** | 6.4 | **45.00** |
| SSIM ↑ | | 0.12 | **0.17** | 0.21 | **0.79** | 0.19 | **0.987** |
| KID ↓ | | **93.2** | 114.0 | 272.5 | **1.2** | 342.0 | **0.1** |
| LPIPS ↓ | | 0.66 | **0.6** | 0.72 | **0.1** | 0.73 | **0.0012** |
| top-1 ↑ | | 1.5 | **7.2** | 4.0 | **68.5** | 6.4 | **75.4** |

fails to inpaint the hard example, and DPS and DDRM inpaint blurry contents. More examples are provided in the appendix.

Quantitative results are also listed in Table 1. One can see that RED-diff consistently outperforms the alternative samplers across all metrics such as KID and PSNR with a significant margin. This indicates not only more faithful restoration by RED-diff but also better perceptual quality images compared with alternative samplers.

Finally, note that RED-diff iterations are quite lightweight with only forward passing to the diffusion score network. In contrast, DPS and ΠGDM require score network inversion by differentiating through the diffusion denoisers. This in turn is a source of instability and renders the steps computationally expensive. Likewise, DDRM involves SVD calculations that are costly. We empirically validate these by comparing the time per-step and GPU memory usage in Table 1.

### 5.2 NONLINEAR INVERSE PROBLEMS

For various nonlinear inverse problems we assess RED-diff on ImageNet data. We choose DPS as the baseline since ΠGDM and DDRM only deal with linear inverse problems.

**High dynamic range (HDR)**. We choose the nolinear HDR task as a candidate to verify RED-diff. HDR performs the clipping function $f(x) = \text{clip}(2x, -1, 1)$ on the normalized RGB pixels. Again, we choose $\lambda = 0.25$ and $lr = 0.5$, and 100 steps. For DPS we choose $\zeta_i = \frac{0.1}{\|y - A(\hat{x}_0(x_i))\|}$ after grid search over the nominator. While RED-diff converges to good solutions, DPS struggles to find a decent solution even after tuning. The metrics listed under Table 2 demonstrate the gap.

**Phase retrieval**. We test on phase retrieval task as well. The task deals with reconstructing the phase from only magnitude observations in the Fourier domain. It is a difficult task especially for ImageNet dataset with diverse details and structures. Again, for RED-diff we use the weight $\lambda = 0.25$ and $lr = 0.5$, while for DPS we optimize the step size $\zeta_i = \frac{0.4}{\|y - A(\hat{x}_0(x_i))\|}$. While both methods face with challenges for recovering faithful images, RED-diff performs better and achieves higher scores for most of the metrics; see Table 2. Note that, phase retrieval from arbitrary measurements is known to be a challenging task. Thus, for a better assessment of RED-diff one can use supported measurement models (e.g., to Gaussian or coded diffraction patterns) that lead to better quality Metzler et al. (2018). That would however need a separate study that we leave for future research.

**Deblurring**. We also test another nonlinear scenario that deals with nonlinear deblurring. We adopt the same setup as in DPS (Chung et al., 2022a) with the blur kernel adopted from a pretrained UNet. For RED-diff we choose $\lambda = 0.25$ and $lr = 0.5$. For DPS also after grid search over the coefficients we end up with $\zeta_i = \frac{1.0}{\|y - A(\hat{x}_0(x_i))\|}$. DPS struggles for this nonlinear tasks. In general, DPS is

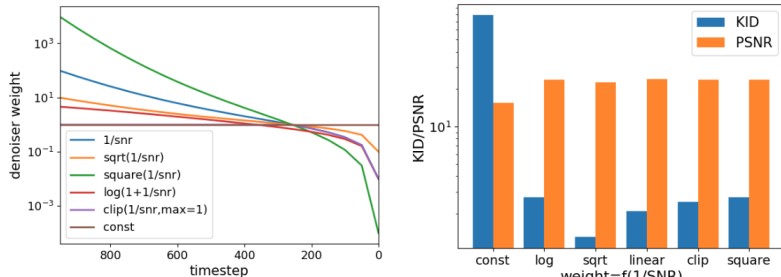

Figure 3: Ablation for denoiser weight tuning. Left: denoiser weight over timesteps (reversed); right: KID and PSNR vs. $\lambda$ for different monotonic functions of inverse SNR.

sensitive to step size and initialization, while RED-diff is not sensitive and achieves much better scores as listed in Table 2.

### 5.3 ABLATIONS

We provide ablations to verify the role of different design components in the proposed algorithm such as denoiser weight tuning, and sampling strategy.

#### 5.3.1 DENOISER WEIGHTING MECHANISM

As discussed in Section 4.2, the variational sampler resembles regularization by the denoising diffusion process. When sampling in descending order, namely from $t = T$ to $t = 1$, each denoiser regularizes different structures from high-level semantics to low-level fine details, respectively. To effect prior at different image scales, each denoiser needs to be tuned properly. We proposed inverse SNR (i.e., $1/\mathrm{SNR}_t$) as the base weight per timestep in Section 4.3. To validate that choice, we ablate different monotonic functions of SNR to weight denoisers over time. The weights are plotted in Fig. 3 (left) over timesteps. The corresponding KID and PSNR metrics are also shown in Fig. 3 (right) for Platte inpainting for different weighting mechanisms. It is observed that the square root decay $(1/\sqrt{\mathrm{SNR}})$ and linear schedule $(1/\mathrm{SNR})$ are the best strategies for KID and PSNR, respectively.

#### 5.3.2 TIMESTEP SAMPLING

We consider five different strategies when sampling the timestep $t$ during optimization, namely: (1) random sampling; (2) ascending; (3) descending; (4) min-batch random sampling; and (5) mini-batch descending sampling. We adopt Adam optimizer with $1,000$ steps and choose the linear weighting mechanism with $\lambda = 0.25$. Random sampling (1) uniformly selects a timestep $t \in [1, T]$, while ascending and descending sampling are ordered over timesteps. It is seen that descending sampling performs significantly better than others. It starts from the denoiser at time $t = T$, adding semantic structures initially, and then fine details are gradually added in the process. This appears to generate images with high fidelity and perceptual quality. We also tested batch sampling with $25$ denoisers per iteration, for $40$ iterations. It is observed that batch sampling smears the fine texture details. See appendix for more details.

## 6 CONCLUSIONS AND LIMITATIONS

This paper focuses on the universal sampling of inverse problems based on diffusion priors. It introduces a variational sampler, termed RED-diff, that naturally promotes regularization by the denoising diffusion process (DDP). Denoisers at different steps of DDP concurrently impose structural constraints from high-level semantics to low-level fine details. To properly tune the regularization, we propose a weighting mechanism based on denoising SNR. Our novel perspective views sampling as stochastic optimization that embraces off-the-shelf optimizers for efficient and tunable sampling. Our experiments for several image restoration tasks exhibit the strong performance of RED-diff compared with state-of-the-art alternatives for inverse problems.

One of the limitations of our variational sampler pertains to the lack of diversity. It is mode-seeking in nature and promotes MAP solutions. We will investigate methods that encourage diversity e.g., by tuning the optimizer, introducing more expressively in the variational distribution, or modifying the criterion by adding dispersion terms as in Stein variational gradient descent Liu & Wang (2016). Additionally, more extensive experiments for 3D generation tasks will solidify the merits of our variational sampler.

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

# A    PROOFS AND DERIVATIONS

## A.1    PROOF OF PROPOSITION 1

As discussed in section 4, using Bayes rule, one can re-write the KL objective in equation 5 as

$$KL\big(q(x_0|y)\|p(x_0|y)\big) = \underbrace{-\mathbb{E}_{q(x_0|y)}\big[\log p(y|x_0)\big] + KL\big(q(x_0|y)\|p(x_0)\big)}_{\text{term(i)}} + \underbrace{\log p(y)}_{\text{term(ii)}}$$

For minimization purposes, term(ii) is a constant, and we can ignore it. The term(i) however has also two parts. The first part is simply the reconstruction loss. Based on the measurement model in equation 2 of the main paper, since we assumed noise is i.i.d. Gaussian, the first part is simply derived as

$$\mathbb{E}_{q(x_0|y)}[\log p(y|x_0)] = -\frac{1}{2\sigma_v^2}\mathbb{E}_{q(x_0|y)}[\|y - f(x_0)\|^2]$$

Using Theorem 2 in Song et al. (2021a), assuming that the score is learned exactly, namely $\epsilon_\theta(x_t; t) = -\sigma_t \nabla_{x_t} \log p(x_t)$, under some mild assumptions on the growth of $\log q(x_t|y)$ and $p(x_t)$ at infinity, we have

$$KL(q(x_0|y)\|p(x_0))) = \int_0^T \frac{\beta(t)}{2}\mathbb{E}_{q(x_t|y)}\Big[\big\|\nabla_{x_t} \log q(x_t|y) - \nabla_{x_t} \log p(x_t)\big\|^2\Big]dt \qquad (12)$$

over the denoising diffusion trajectory $\{x_t\}$ for positive values $\{\beta(t)\}$. This essentially implies that a weighted score-matching over the continuous denoising diffusion trajectory is equal to the KL divergence. In practice, however, we are often interested in a reweighting of r.h.s. in Eq. 12 that leads to other divergence measures Song et al. (2021a).

## A.2    PROOF OF PROPOSITION 2

The regularization term is essentially the score matching loss in the r.h.s. of equation 12. In practice, we often use a weighting scheme different than $\beta(t)/2$ that corresponds to maximum likelihood estimation Song et al. (2021a). For re-weighting, it is useful to recognize the following Lemma, which leverages integration by part, with the complete proof provided in Song et al. (2021a).

**Lemma 1**. *The time-derivative of the KL divergence at timestep $t$ obeys*

$$\frac{dKL\big(q(x_t|y)\|p(x_t)\big)}{dt} = -\frac{\beta(t)}{2}\mathbb{E}_{q(x_t|y)}\Big[\big\|\nabla_{x_t} \log q(x_t|y) - \nabla_{x_t} \log p(x_t)\big\|^2\Big] \qquad (13)$$

This Lemma is intuitive based on equation 12. Now, to allow a generic weighting, one can simply weight equation 13 with $\omega(t)$, Then, under the condition $\omega(0) = 0$, the r.h.s. of equation 12 can be written as (see e.g., Song et al. (2021a))

$$\int_0^T \frac{\beta(t)}{2}\omega(t)\mathbb{E}_{q(x_t|y)}\Big[\big\|\nabla_{x_t} \log q(x_t|y) - \nabla_{x_t} \log p(x_t)\big\|^2\Big]dt$$

$$= -\int_0^T \omega(t)\frac{dKL\big(q(x_t|y)\|p(x_t)\big)}{dt}dt$$

$$\stackrel{(a)}{=} \underbrace{-\omega(t)KL\big(q(x_t|y)\|p(x_t)\big)\Big]_0^T}_{=0} + \int_0^T \omega'(t)KL\big(q(x_t|y)\|p(x_t)\big)dt$$

$$= \int_0^T \omega'(t)\mathbb{E}_{q(x_t|y)}\Big[\log \frac{q(x_t|y)}{p(x_t)}\Big]dt$$

where $\omega'(t) := \frac{d\omega(t)}{dt}$. The equality in (a) holds because $\omega(t)KL\big(q(x_t|y)\|p(x_t)\big)\Big]_0^T$ is zero at $t = 0$ and $t = T$. This is because $\omega(t) = 0$ by assumption at $t = 0$, and $x_T$ becomes a pure Gaussian noise at the end of the diffusion process which makes $p(x_T) = q(x_T|y)$ and thus $KL\big(q(x_T|y)\|p(x_T)\big) = 0$.

For simplicity, let us suppose $\sigma = 0$ in the variational distribution, and thus $x_0 = \mu$ is deterministic. The forward diffusion process is then $x_t = \alpha_t \mu + \sigma_t \epsilon$ for $\epsilon \sim \mathcal{N}(0, 1)$. Applying the re-parameterization trick, the gradient w.r.t $\mu$ can be simply written as

$$\nabla_\mu \mathrm{reg}(\mu) = 2\sigma_v^2 \int_0^T \omega'(t) \mathbb{E}_{\epsilon \sim \mathcal{N}(0,1)} \Big[ \big( \nabla_{x_t} \log q_t(x_t|y) - \nabla_{x_t} \log p(x_t) \big)^\top \frac{dx_t}{d\mu} \Big] dt$$

$$= 2\sigma_v^2 \int_0^T \omega'(t) \mathbb{E}_{\epsilon \sim \mathcal{N}(0,1)} \Big[ \big( (-\frac{\epsilon}{\sigma_t}) - (-\frac{\epsilon_\theta(x_t; t)}{\sigma_t}) \big)^\top (\alpha_t I) \Big] dt$$

where $q(x_t|y) = \mathcal{N}(\alpha_t\mu, \sigma_t^2 I)$, and $\nabla_{x_t} \log q_t(x_t|y) = -(x_t - \alpha_t\mu)/\sigma_t^2 = -\epsilon/\sigma_t$. Note, the gradient was exchanged with the expectation since both gradient terms exist and are bounded. Finally, we can rearrange terms to arrive at the compact form

$$\nabla_\mu \mathrm{reg}(\mu) = \frac{1}{T} \int_0^T T\omega'(t) \frac{2\sigma_v^2 \alpha_t}{\sigma_t} \mathbb{E}_{\epsilon \sim \mathcal{N}(0,1)} \big[ (\epsilon_\theta(x_t; t) - \epsilon) \big] dt$$

$$= \mathbb{E}_{t \sim \mathcal{U}[0,T], \epsilon \sim \mathcal{N}(0,1)} \big[ \lambda_t (\epsilon_\theta(x_t; t) - \epsilon) \big]$$

for $\lambda_t := T\omega'(t) 2\sigma_v^2 \alpha_t / \sigma_t$. Note that we can ignore the second term inside the expectation since $\epsilon$ has a zero mean.

### A.3 Adding dispersion to variational approximation

The results in Proposition 2 are presented for Dirac distribution with no dispersion, namely $q(x_0|y) \sim \delta(x_0 - \mu)$. We however can easily extend those to optimize for Gaussian dispersion as well. The interesting observation is that the gradient w.r.t. the mean $\mu$ remains the same, and the gradient w.r.t. $\sigma$ is simple and tractable.

In this case, we have $q(x_0|y) \sim \mathcal{N}(\mu, \sigma^2 I)$. The diffusion signal at timestep $t$ can then be represented in a compact form using reparameterization trick as $x_t = \alpha_t \mu + \sqrt{\alpha_t^2 \sigma^2 + \sigma_t^2} \epsilon$. Let's define $\eta_t := (1 + \sigma^2 (\frac{\alpha_t}{\sigma_t})^2)^{1/2}$ so that $x_t = \alpha_t \mu + \eta_t \sigma_t \epsilon$. Note, for no dispersion case $\eta_t = 1$. Then, gradient w.r.t. the mean is obtained as

$$\nabla_\mu \mathrm{reg}(\mu, \sigma) = \int_0^T \omega'(t) \mathbb{E}_{\epsilon \sim \mathcal{N}(0,1)} \Big[ \big( \nabla_{x_t} \log q_t(x_t|y) - \nabla_{x_t} \log p(x_t) \big)^\top \frac{dx_t}{d\mu} \Big] dt$$

$$= \int_0^T \omega'(t) \mathbb{E}_{\epsilon \sim \mathcal{N}(0,1)} \Big[ \big( (-\frac{\epsilon}{\sqrt{\alpha_t \sigma^2 + \sigma_t^2}}) - (-\frac{\epsilon_\theta(x_t; t)}{\sigma_t}) \big)^\top (\alpha_t I) \Big] dt$$

$$= \int_0^T \omega'(t) \mathbb{E}_{\epsilon \sim \mathcal{N}(0,1)} \frac{\alpha_t}{\sigma_t} \Big[ \epsilon_\theta(x_t; t) - \eta_t^{-1} \epsilon \Big] dt$$

$$= \mathbb{E}_{\epsilon, t} \big[ \lambda_t \epsilon_\theta(x_t; t) \big]$$

Similarly, the gradient w.r.t. the dispersion is found as

$$\nabla_\sigma \mathrm{reg}(\mu, \sigma) = \int_0^T \omega'(t) \mathbb{E}_{\epsilon \sim \mathcal{N}(0,1)} \Big[ \big( \nabla_{x_t} \log q_t(x_t|y) - \nabla_{x_t} \log p(x_t) \big)^\top \frac{dx_t}{d\sigma} \Big] dt$$

$$= \int_0^T \omega'(t) \mathbb{E}_{\epsilon \sim \mathcal{N}(0,1)} \Big[ \big( (-\frac{\epsilon}{\sqrt{\alpha_t \sigma^2 + \sigma_t^2}}) - (-\frac{\epsilon_\theta(x_t; t)}{\sigma_t}) \big)^\top \epsilon (\alpha_t \sigma^2 + \sigma_t^2)^{-1/2} 2\alpha_t^2 \sigma \Big] dt$$

$$= \sigma \mathbb{E}_{\epsilon, t} \Big[ \lambda_t 2\eta_t^{-1} (\frac{\alpha_t}{\sigma_t}) \epsilon^\top (\epsilon_\theta(x_t; t) - \eta_t^{-1} \epsilon) \Big]$$

The dispersion gradient has close form. However, we believe the Gaussian dispersion is not the right choice to add stochasticity and diversity to natural images since simply perturbing an image with Gaussian noise does not lead to another legitimate image. In essence, one needs a proper dispersion on the image manifold that needs more sophisticated dispersion models such as in Wang et al. (2023).

Finally, it is also useful to note that we believe our framework can be extended to more complex variational distributions such as hierarchical, multimodal, or normalizing flow-based distributions as in VAEs, following the recipe in the previous work such as Vahdat et al. (2021).

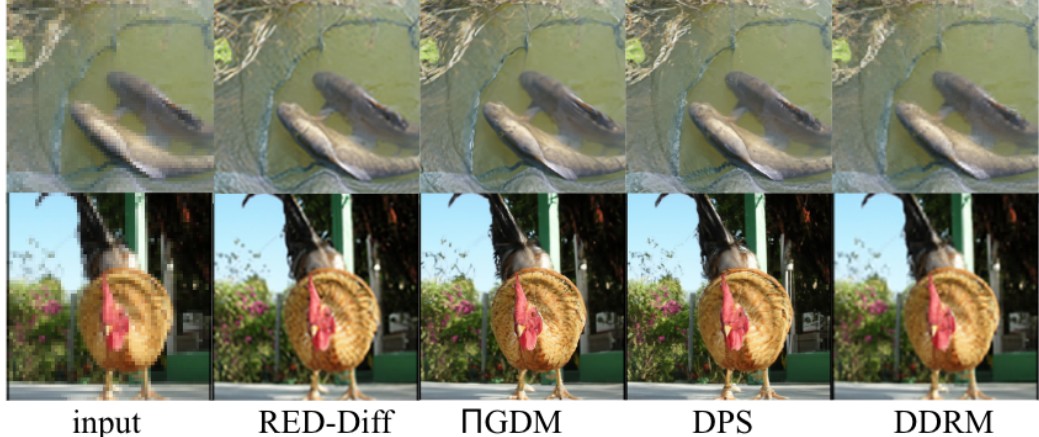

Figure 4: Comparison of the proposed variational sampler with alternatives for superresolution of representative ImageNet examples. Each sampler is tuned for the best performance.

Table 3: Performance of different samplers for ImageNet 4x superresolution. We adopt unconditional guided diffusion model for the score function. We choose Adam $lr = 0.25$.

| Sampler | PSNR(dB) ↑ | SSIM ↑ | KID ↓ | LPIPS ↓ | top-1 ↑ |
|---|---|---|---|---|---|
| DPS | 24.83 | 0.71 | 10.01 | 0.16 | **71.5** |
| ΠGDM | 25.25 | 0.73 | 10.9 | **0.15** | 71.02 |
| DDRM | 25.32 | 0.72 | 14.0 | 0.23 | 63.9 |
| RED-diff | **25.95** | **0.75** | **10.0** | 0.25 | 66.7 |

## B    ADDITIONAL EXPERIMENTS

### B.1    PRETRAINED DIFFUSION MODEL

We adopt the score function from the pretrained guided diffusion model, which uses no class conditioning. It is trained on $256 \times 256$ ImageNet dataset. Architecture details are listed in section 3 of Dhariwal & Nichol (2021).

### B.2    IMAGE SUPERRESOLUTION

We perform 4x superresolution on a 1k subset of ImageNet dataset. Bicubic degradation is applied to create the low-resolution inputs. After tuning $\lambda = 0.25$, Adam iterations are run for $1,000$ steps. A few samples are illustrated in Fig. 4, where one can notice that RED-diff strikes a good balance between image fidelity and perceptual quality. One can play with this trade-off by tuning the Adam learning rate. Smaller learning rates lead to higher fidelity, while larger learning rates give rise to higher perceptual quality. See supplementary material for more examples.

Quantitative results are also listed in Table 3. One can see that RED-diff significantly outperforms the alternative samplers in terms of PSNR and SSIM. Table 3 however indicates that RED-diff is not as good as other alternatives in terms of the perceptual quality. We want to note the trade off we observe between fidelity (e.g., PSNR) and perceptual quality (e.g., LPIPS). It seems that one can achieve better perceptual quality at the expense of lower fidelity by tuning the regurlization weight $\lambda$ that control the bias-variance trade off. Other hyperparamaters such as step size and number of steps can be tuned alternatively. Fig. 5 depicts the trade off between PSNR-LPIPS, where the red-cross denotes the point reported in Table 3.

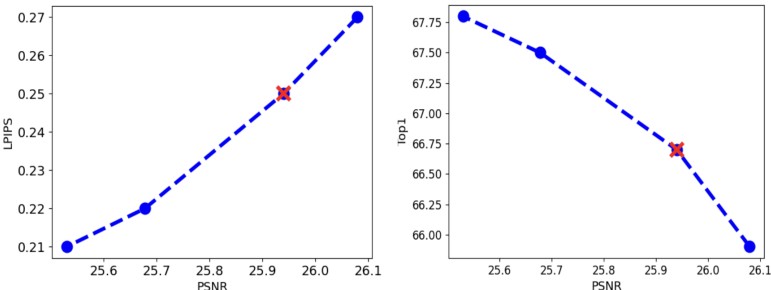

Figure 5: LPIPS vs. PSNR (dB) trade-off for 4x superresolution.

### B.3 MORE EXAMPLES FOR COMPARISONS

We provide additional examples to compare RED-diff with alternative methods for inpainting and superresolution. The inpainting examples are shown in Fig. 6. Superresolution examples are also shown in Fig. 7. For both tasks, the same setup as discussed n the main paper was used for each scheme.

### B.4 DIVERSITY

To verify the sample diversity for RED-diff, Fig. 8 illustrates different samples for the ImageNet inpainting task when the seed for $\epsilon$ is changing. We choose Adam optimizer with $lr = 0.25$, and $1,000$ steps. From the examples, we observe that RED-diff samples are sufficiently diverse. To further enhance the diversity, one can play with the optimizer parameters, for example by choosing a larger learning rate for Adam, or using a smaller number of steps.

### B.5 DIFFUSION EVOLUTION

To understand the restoration process with diffusions, we plot the evolution of the diffusion model over timesteps in Fig. 9. We visualize $\hat{\mu}$ as the outputs from every $100$ steps over $1,000$ timesteps. To gain further insights into the generation process and how the image structures are generated, the evolution of diffusion steps is also plotted in the frequency domain. Fig. 10 illustrates the magnitude and phase every $100$ steps over $1,000$ timesteps. It can be observed that the earlier denoisers add high frequency details, while the later denoisers generate low frequency structures.

## C ADDITIONAL SCENARIOS

We include additional tasks to assess RED-diff for noisy inverse problems. We consider noisy inpainting and compressed sensing MRI.

### C.1 COMPRESSED SENSING MRI

We adopt the pretrained diffusion model from Jalal et al. (2021), and sample via RED-Diff. The pretrained diffusion has been trained based on fastMRI brain data. We test sampling for both in-domain brain, and out-of-domain knee data. The performance is compared with CSGM-Langevin Jalal et al. (2021) for which the codebase is publicly available, and serves as a state-of-the-art for complex-valued medical image reconstruction. We use the multi-coil fastMRI brain dataset Zbontar et al. (2018) with 1D equispaced undersampling, and the fully-sampled 3D fast-spin echo multi-coil knee MRI dataset from Ong et al. (2018) with 2D Poisson Disc undersampling mask, as in Jalal et al. (2021). We used 6 validation volumes for fastMRI, and 3 volumes for Mridata by selecting 32 middle slices from each volume. We use exactly the same tuning parameters as for the inpainting task. The results for different undersampling rates (R) are shown in Table 4. Obviously, RED-Diff outperforms CSGM method (with Langevin sampling) consistently in terms of PSNR, that is the measure of interest in MRI reconstruction. For more details see Ozturkler et al. (2023). Representative

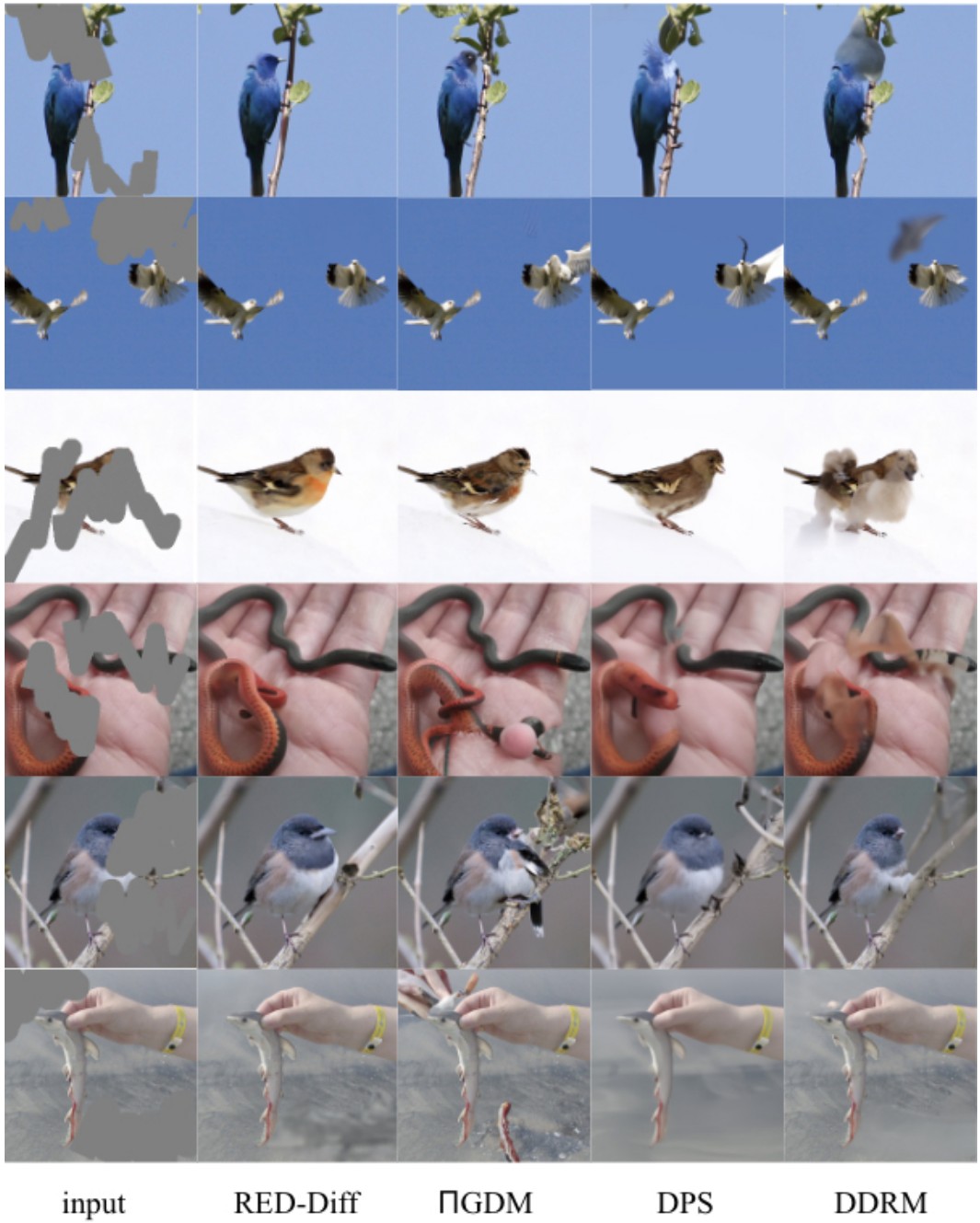

input      RED-Diff      ΠGDM      DPS      DDRM

Figure 6: Comparison of the proposed variational sampler with alternatives for inpainting representative ImageNet examples. Each sampler is tuned for the best performance.

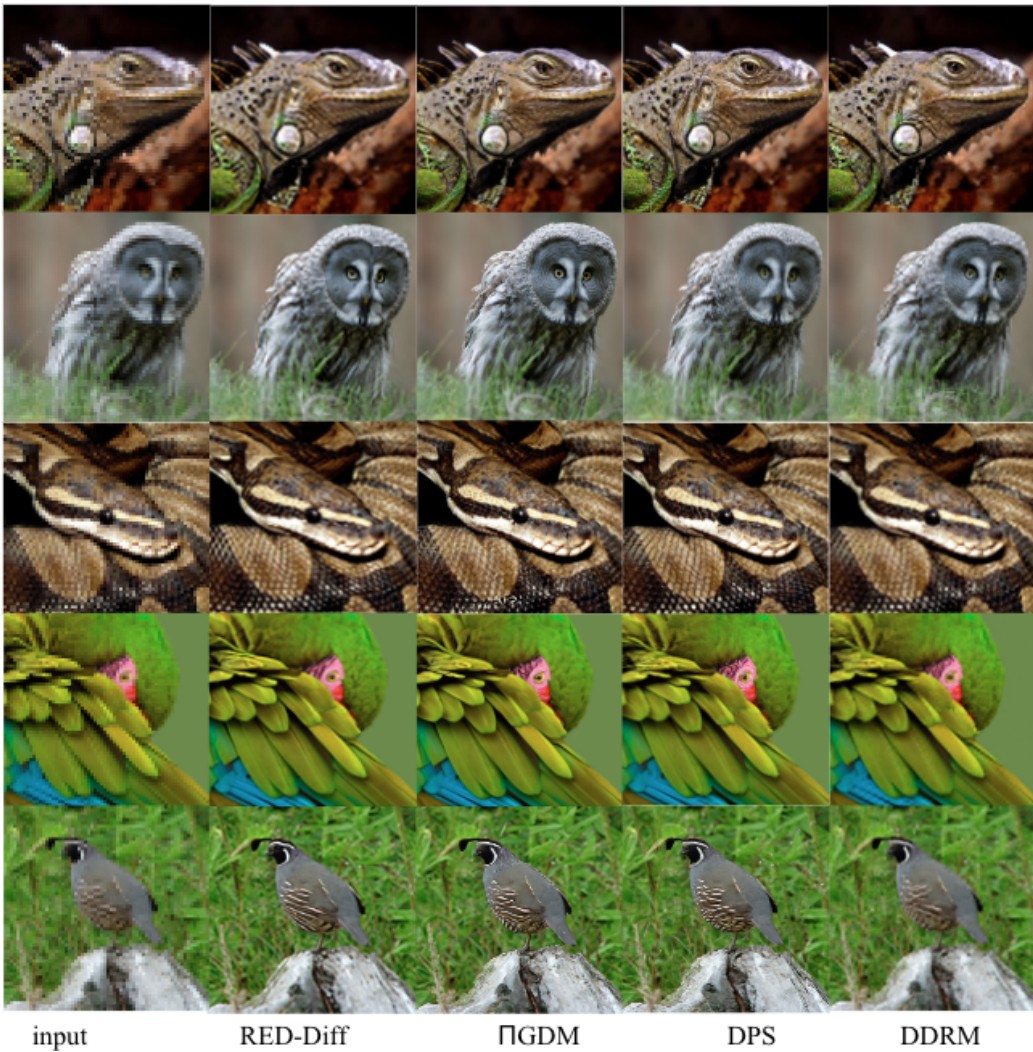

Figure 7: Comparison of the proposed variational sampler with alternatives for superresolution of representative ImageNet examples. Each sampler is tuned for the best performance.

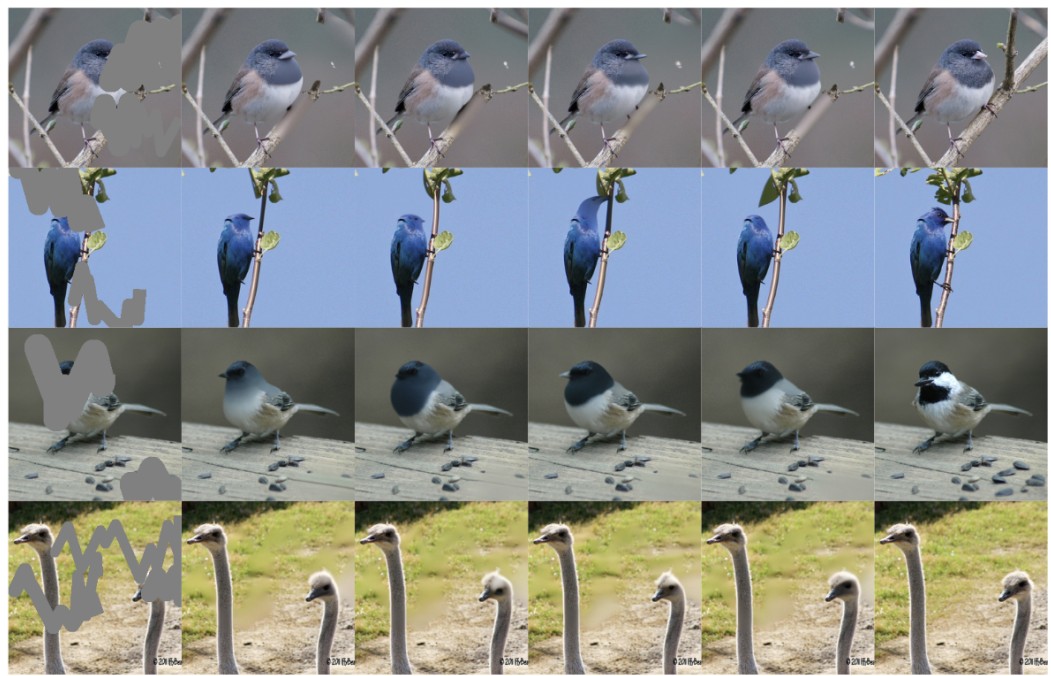

Figure 8: Sampling diversity for ImageNet inpainting. A few random realizations are shown, where the first column is the masked input image, and the last one is the reference image.

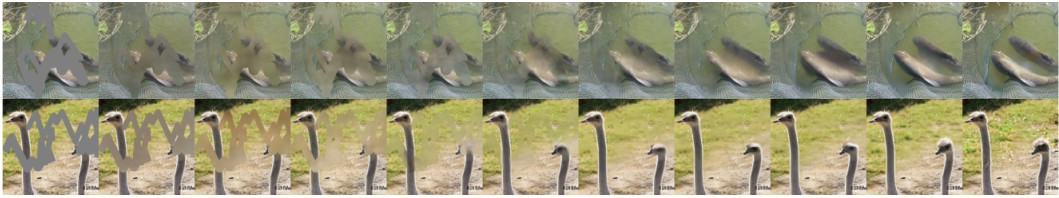

Figure 9: Evolution of RED-diff over iterations. Descending sampling direction is used. Denoisers from large to small $t$ restore high-level to low-level features, respectively. Adam with $1,000$ steps used and every $100$ iterations are visualized.

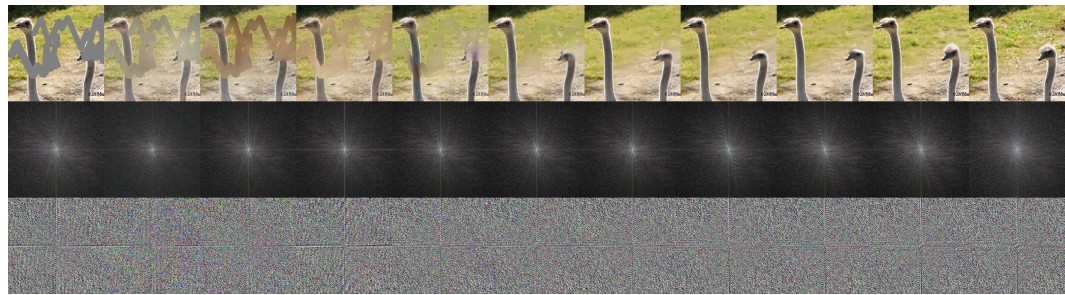

Figure 10: Evolution of RED-diff over iterations in the frequency domain. Middle row shows the log-magnitude, and the bottom row shows the phase.

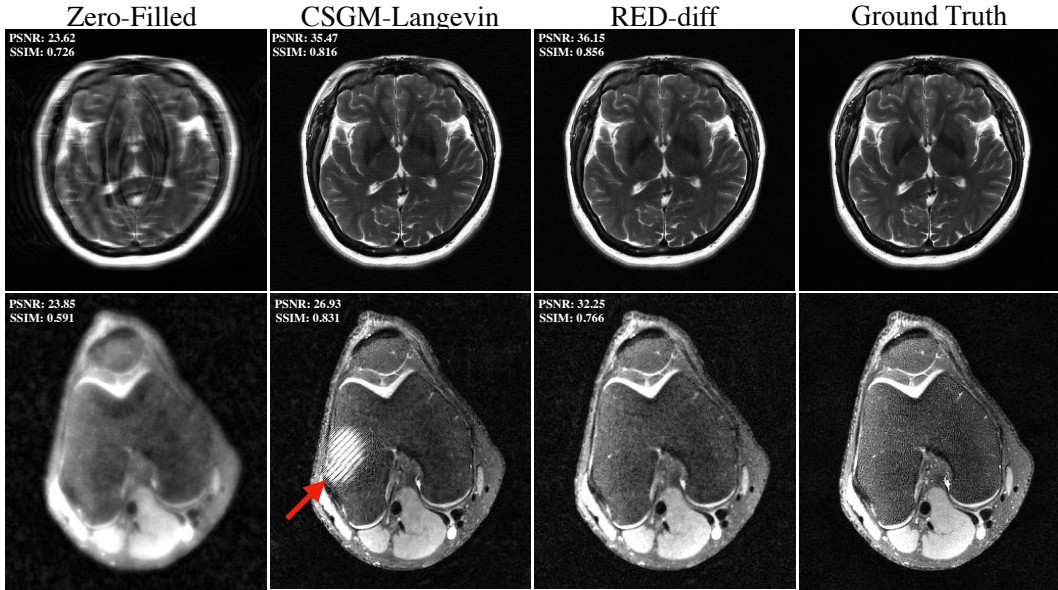

Figure 11: Representative reconstructed images for brain with $R = 4$, and knee with $R = 12$.

Table 4: Compressed sensing MRI PSNR (dB) for fastMRI brain and Mridata knee dataset with Rx undersampling. CSGM is a SOTA method for complex-valued MRI with a public codebase.

| Anatomy | Brain | Knee | | time |
|---|---|---|---|---|
| Sampler | $R = 4$ | $R = 12$ | $R = 16$ | (sec/iter) |
| CSGM | 36.3 | 31.4 | 31.8 | 0.344 |
| RED-diff | **37.1** | **33.2** | **32.7** | **0.114** |

samples are also shown in Fig. 11. At this point, it is also worth noting that RED-diff relates to other variational approaches in the imaging literature such as Knoll et al. (2011); Portilla et al. (2003); Kobler et al. (2017). A through comparison is needed and that is left for future research.

## C.2 NOISY INPAINTING

To see the effects of measurement noise on RED-diff performance, we add Gaussian noise with $\sigma_v = 0.1$ to the masked Palette images for inpainting. We compare RED-diff with DPS and PGDM, where for all we use 100 steps. For RED-diff we choose $\lambda = 0.25$ similar to all other scenarios with $lr = 0.25$. For DPS, we choose $\eta = 0.5$, and the step size run $\zeta = 0.5/\|y - A(\hat{x}_0(x))\|$ adopted from Chung et al. (2022a) where we run a grid search over the range $[0, 1]$ for the coeffficient. Table 5 shows the metrics. It is seen that RED-diff outperforms DPS and $Pi$GDM in most metrics.

Table 5: Noisy inpainting with $\sigma_y = 0.1$ for Palette data with 100 steps.

| Sampler | PSNR(dB) ↑ | SSIM ↑ | KID ↓ | LPIPS ↓ | top-1 ↑ |
|---|---|---|---|---|---|
| DPS | 16.98 | 0.39 | **1.9** | 0.52 | 8.8 |
| ΠGDM | 16.15 | 0.31 | 3.6 | 0.45 | 42.4 |
| RED-diff | **18.92** | **0.41** | 3.4 | **0.4** | **43.3** |

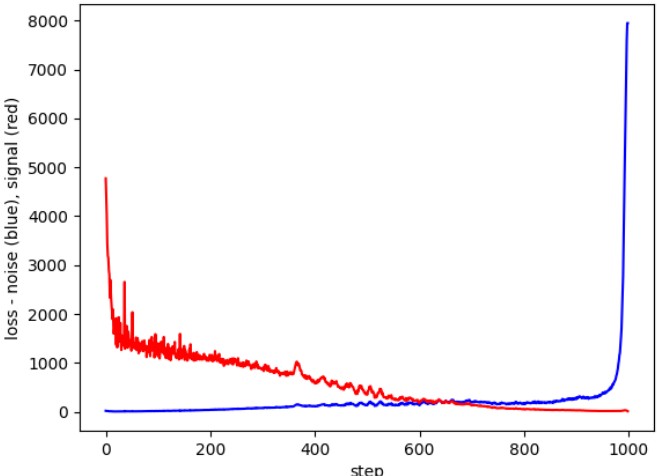

Figure 12: Evolution of signal residual $\|\hat{\mu}_t - \mu\|^2$ (red) and noise residual $\|\epsilon_\theta(x_t; t) - \epsilon\|^2$ (blue) over time for ImageNet inpainting when equally weighting diffusion denoisers at different timesteps.

## D  ABLATIONS

### D.1  DENOISER WEIGHTING MECHANISM

It was discussed in section 4.3 that the noise residual blows up at the last timesteps of the diffusion process. Here, we illustrate that in Fig. 12 when denoisers are equally weighted. It shows the magnitude of both the signal and noise residuals. It is clearly seen that the noise residual goes up drastically at around $t = 1,000$. Similarly, the signal residual also blows up at earlier steps. This suggests a mechanism that guarantees a relatively small signal residual at all iterations that led to the SNR ruler for weighting the denoisers.

### D.2  OPTIMIZING RED-DIFF FOR MORE EPOCHS

Since RED-diff is an optimization-based sampling, one may ponder that using more iterations by going over the denoisers more than once can improve the sample quality. To test this idea, we choose different epochs for ImageNet inpainting, when we use Adam optimizer with $1,000$ steps per epoch. A few representative examples are shown in Fig. 13. It is observed that adding more epochs has negligible improvement on the sample quality.

### D.3  OPTIMIZATION STRATEGY

The proposed variational sampler relies on optimization for sampling. To see the role of optimizer, we first ablate SGD versus Adam. We found SGD more sensitive to step size which in turn demands more careful tuning. We tune the hyperparameters in each case for the best performance for ImageNet inpainting with $\lambda = 0.25$. We use Adam with parameters discussed in the beginning of Section 5. Fig. 14 shows representative inpainting examples, where SGD is tested with momentum $(0.9)$ and without momentum. It appears that SGD with momentum can be as good as the Adam optimizer, which indicates that RED-diff is not sensitive to the choice of optimizer. Our future direction will explore accelerated optimization for faster convergence.

For the Adam optimizer, we also ablated the initial learning rate to see its effect on the quality of the generated samples. The results are depicted in Fig. 15 (right). It appears that tuning Adam learning rate is important, where a smaller learning rate (e.g., $0.05$) leads to better reconstruction quality since the optimizer can better converge to the (optimal) MAP estimate. However, using a larger learning rate (e.g., $0.5$) leads to better perceptual quality measured with KID.

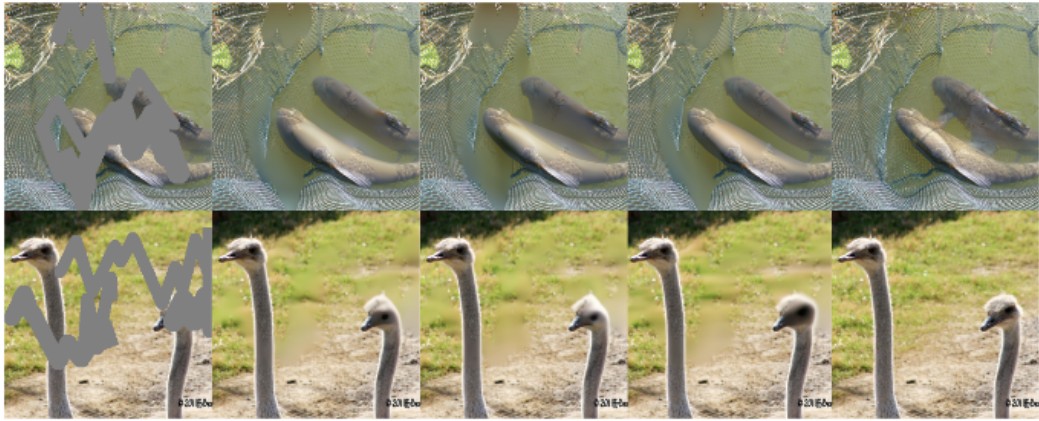

Figure 13: RED-diff outputs for a different number of epochs. From left to right: input, epochs=1,2,10, and the reference, respectively. Adding more epochs does not make any noticeable difference.

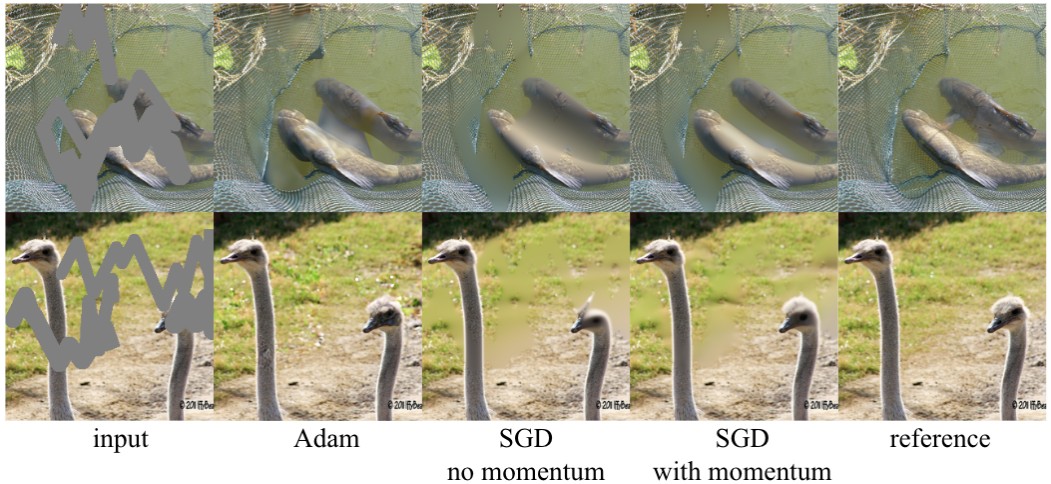

Figure 14: Ablation of optimization strategy for an inpainting example from ImageNet dataset. Adam and SGD (with momentum 0.9) seem to perform similarly.

KID for a range of $\lambda$ is also depicted in Fig. 15 (left). It is evident that there is an optimal $\lambda$, which confirms that with proper tuning of the bias-variance trade-off one can gauge the sampling quality.

### D.4  TIMESTEP SAMPLING STRATEGY

We ablate the timestep sampling strategy and the number of sampling steps. For comparison, we consider five different strategies when sampling from $t$ during optimization. This includes (1) random sampling; (2) ascending; (3) descending; (4) min-batch random sampling; and (5) mini-batch descending sampling. We adopt Adam optimizer with $1,000$ steps and choose the linear weighting mechanism with $\lambda = 0.25$. Random sampling (1) uniformly selects a timestep $t \in [1, T]$, while ascending and descending sampling are ordered over timesteps. As shown in the inpainting example in Fig. 16, it is seen that descending sampling performs significantly better than others. It starts from the denoiser at time $t = T$, adding semantic structures initially, and then fine details are gradually added in the process. This appears to generate images with high fidelity and perceptual quality.

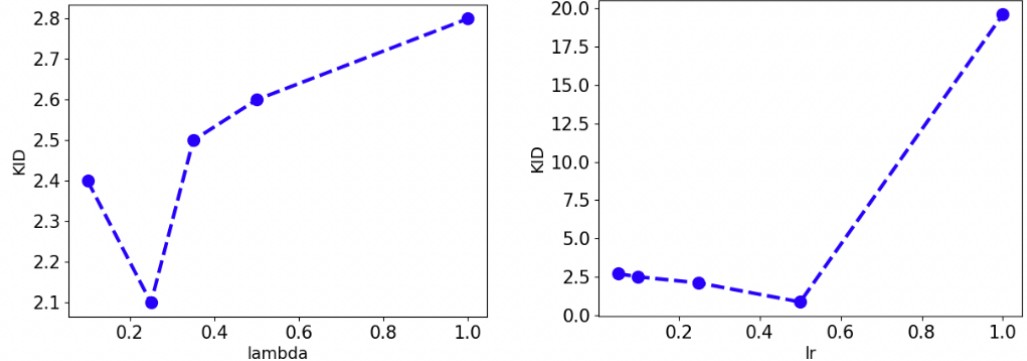

Figure 15: KID versus the optimizer learning rate (right), and the weight $\lambda$ (left).

The aforementioned sampling strategies choose a single denoiser at a time, but one may ponder what if one performs batch sampling to regularize based on multiple denoisers at the same time. This is also computationally appealing as it can be executed in parallel based on our optimization framework. To test this idea, we sort $1,000$ time steps in descending order, and use a batch of 25 denoisers per iteration, for 40 iterations. It is again observed from Fig. 16 that batch sampling smears the fine texture details. This needs further investigation to find out the proper weighting mechanism that enables parallel sampling. In conclusion, descending sampling with a single (or a few) denoiser at a time seems to be the best sampling strategy. More details are found in the supplementary material.

Table 6: Performance of RED-diff for inpainting under different step counts, when $\lambda = 0.25$ and $lr = 0.25$.

| steps | PSNR(dB) ↑ | SSIM ↑ | KID ↓ | LPIPS ↓ | top-1 ↑ |
|---|---|---|---|---|---|
| 10 | 18.83 | 0.75 | 22.05 | 0.21 | 59.4 |
| 100 | 23.13 | 0.87 | **1.93** | 0.12 | **70.8** |
| 1000 | 23.86 | 0.88 | 2.17 | 0.11 | 69.8 |

It is also useful to understand how many steps RED-diff needs to generate good samples. To this end, we evaluated ImageNet inpainting for a different number of steps in Table 6. One can observe that with 100 steps the best perceptual quality is achieved where KID=1.93. With more steps, the optimizer tends to refine the fidelity so a better reconstruction PSNR is achieved. This ablation suggests that not many steps are needed for the variational sampler. This is in contrast with DreamFusion Poole et al. (2022) which uses a large number of iterations (15k) to generate samples.

### D.5 CONNECTION AND DIFFERENCES WITH RED

We want to further elaborate the connections with the RED framework. It is useful to recognize that the classical RED adds no noise to the input for denoising, and it is simply a fixed-point problem. RED also uses a single *deterministic* denoiser. RED-diff is however fundamentally different. It is generative and add noise to the input of all denoisers in the diffusion trajectory. As a result it stochastically navigates towards the prior. This seems crucial to find a plausible solution. To highlight of the strength of the RED-diff regulrization over RED we performed two simple experiments. In the first experiment, we removed noise from the input of all denosiers, namely $x_t = \mu_t$, and then ran RED-diff iterations for the image inpainting task. It appears that iterations do not progress and they get stuck around the initial masked image as seen in Fig. 17. In the second experiment, we just used a single denoiser at time $t = 0$, which has a very small noise, and thus resembles RED the most. We observe again that the solution gets stuck and cannot navigate to the real region of prior to fill out the masked areas in the image. We tested for a single denoiser picked from other time steps such as $t = 500$, and observed the same behavior.

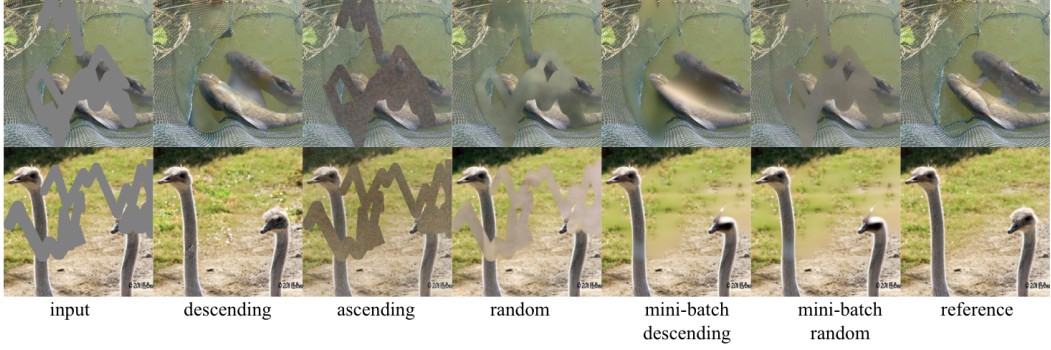

Figure 16: Ablation of sampling strategy for ImageNet inpainting. For mini-batch sampling, the batch size is 25. Sampling in descending direction performs significantly better than the rest. Adam optimizer is used with an unconditional ImageNet guided diffusion score.

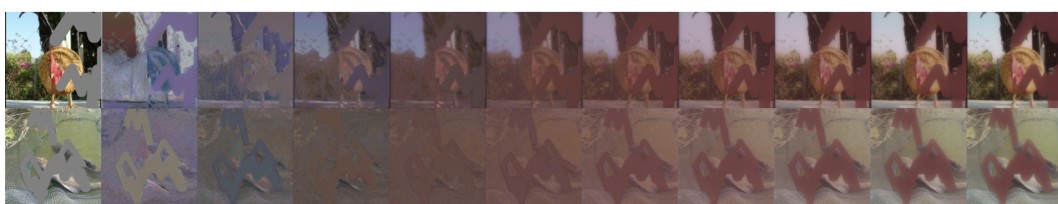

Figure 17: Evolution of diffusion when the noise to each denoiser is removed and $x_t = \mu$.

### D.6 COMPARISON WITH PLUG-AND-PLAY (PNP) METHODS

Plug-and-Play (PnP) methods are an important class of techniques for solving inverse problems; see e.g., Kamilov et al. (2023); Zhang et al. (2021;?); Wei et al. (2022); Laumont et al. (2022); Hurault et al. (2021); Xu et al. (2020); Pesquet et al. (2021). PnP and RED methods are indeed related and connected through fixed-point iterations Cohen et al. (2021). It thus deserves to compare RED-diff with PnP framework. To demonstrate the benefits of RED-diff with respect to PnP methods we compare with two representatives namely DPIR Zhang et al. (2021) and DiffPIR Zhu et al. (2023). DPIR integrates deep denoiser prior into PnP methods. In essence, it first trains a CNN for denosing. It then plugs the deep denoiser prior as a modular part into a half quadratic splitting iterative algorithm for solving inverse problems.

We consider the inpainting task and use the code from public repository DPIR [4] with IRCNN denoiser. To be consistent with the experiments in the paper, we use ImageNet data and random Palette masks as described in section 5.1 of the main paper. For 10 random samples, we compare PSNR and LPIPS in Table 7, and show representative samples in Fig. 18. It is evident that RED-diff achieves a much better quality images thanks to the diffusion prior. Note that DPIR is a purely deterministic method.

For completeness, we also compare with DiffPIR Zhu et al. (2023). It is worth noting that the DiffPIR paper Zhu et al. (2023) has appeared after our initial arXiv submission. For DiffPIR we adopt the code from the public repository [5] with unconditional guided diffusion score function. The results are again shown in Table 7, and representative samples in Fig. 18. RED-diff performs slightly better than DiffPIR Zhu et al. (2023). One however should note that RED-diff provides a rigorous treament from the maximum likelihood perspective (and MAP estimator for no dispersion case) as it directly minimizes the KL divergence, and as a result it better lends itself optimization and tuning. In essence, the optimization step in Algorithm 1 can be replaced with any off-the-shelf optimizer. With the SNR weighting mechanism described in section 4.3, we observe better quality images compared with DiffPIR; see Fig 18 and Table 7. A more through analysis and comparison is left for future research.

---

[4] https://github.com/cszn/DPIR
[5] https://github.com/yuanzhi-zhu/DiffPIR

Table 7: Comparison with plug-and-play prior methods for image inpainting for samples from Imagenet data. RED-diff and DiffPIR use 100 diffusion timesteps and both use unconditional guided diffusion score function.

| Sampler | PSNR(dB) ↑ | SSIM ↑ | KID ↓ | LPIPS ↓ | top-1 ↑ |
|---|---|---|---|---|---|
| DPIR | 16.50 | 0.792 | 42.62 | 0.237 | 60.0 |
| DiffPIR | **27.29** | 0.8728 | 7.96 | 0.1401 | 70.0 |
| RED-diff | 26.06 | **0.8792** | **7.92** | **0.1060** | **90.0** |

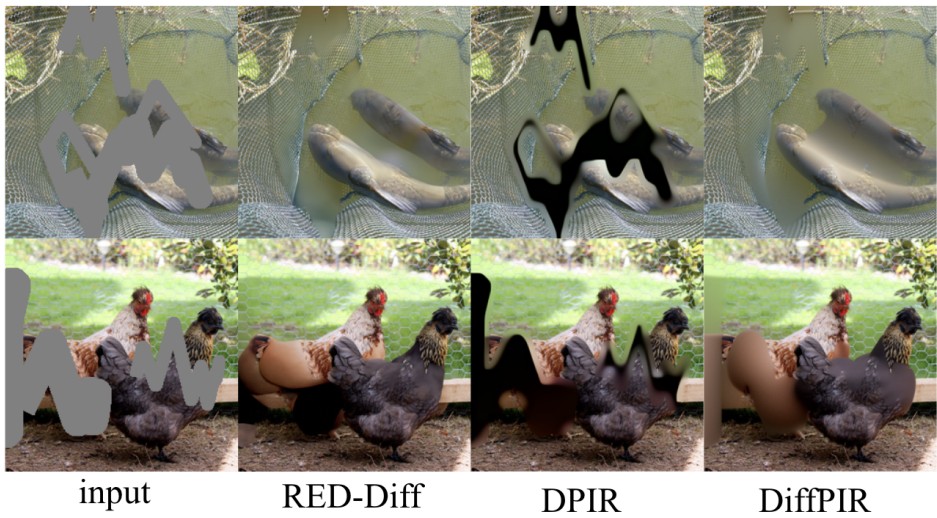

input    RED-Diff    DPIR    DiffPIR

Figure 18: Comparison between RED-diff and plug-play-methods, namely DPIR and DiffPIR for Palette inpainting.

### D.7 EXTENSION TO OBSERVATIONS WITH POISSON NOISE

We used Gaussian noise for the observation model in 2 to simplify the exposition. In essence, the variational formulation is flexible with the choice of the observation noise as it eventually reflects in the reconstruction loss in 6, namely $-\mathbb{E}_{q(x_0|y)}\big[\log p(y|x_0)\big]$. Here, we extend the reconstruction loss to Poisson noise distribution, namely $y \sim \text{Poisson}(\lambda = f(x_0))$, with discrete and positive values such as photon counts in imaging. Suppose that $[f(\mu)]_j > 0$ for all $j$. Then, assuming i.i.d. Poisson noise,

$$p(y|x_0) = \prod_{j=1}^{n} \frac{[f(x_0)]_j^{y_j} \exp([-f(x_0)]_j)}{y_j!} \tag{14}$$

As a result

$$\log p(y|x_0) = \sum_{j=1}^{n} y_j \log[f(x_0)]_j - [f(x_0)]_j - \log y_j! \tag{15}$$

Assuming Dirac distribution for $q(x_0|y) = \mathcal{N}(\mu, \sigma^2)$ with $\sigma \to 0$, the reconstruction loss admits a simple form

$$-\mathbb{E}_{q(x_0|y)}\big[\log p(y|x_0)\big] = \sum_{j=1}^{n} y_j - \log[f(\mu)]_j + [f(\mu)]_j + \log y_j! \tag{16}$$

where the last term in constant. With this loss expression, one can apply (sub)gradient based algorithms to solve RED-diff. It is important to recognize that in contrast with DPS, there is no need for intractable likelihood $\log p(y|x_t)$. As a result, there is no need to for Gaussian approximation of Poisson distribution that limits DPS to work only when the signal intensity is large. This indeed shows another benefit of our variational approximation over DPS.

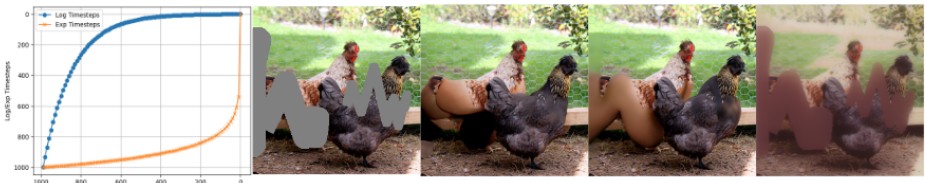

Figure 19: RED-diff under different timestepping strategies. From left to right: timestepping schedule, masked image input, linear timestepping, logarithmic timestepping, and exponential timestepping output.

### D.8 TIMESTEPPING EXPLORATION

Since the ascending sampling works best among the sampling strategies in Fig. xx, we tested ascending sampling with nonuniform timestepping to see if it can reduce the number of steps. We test logarithmic and exponential timestepping. For the discrete timesteps $t \in [1, 1, 000]$, we choose 100 timsteps with logarithmic and exponential spacing; see Fig. 19 (left). We test RED-diff for inpainting task. The sample results are shown in Fig. 19, and compered with uniform spacing. It appears that uniform spacing works the best, and then the logarithmic spacing. This indicates that it is important to sufficiently sample low noise denoisers to recover fine details.

### D.9 CLARIFICATIONS ON THE PROS AND CONS OF VARIATIONAL APPROXIMATION

As it was stated in the main paper, we introduced variational approximation to address the caveats of PGDM and DPS that approximates the *likelihood* with Gaussian distribution around the MMSE estimate. We should note that our variational method is also a unimodal approximation, but it approximates the *posterior* with Gaussian distribution. The benefits of posterior approximation are however threefold: $(i)$ it requires no score Jacobian as opposed to DPS and PGDM that need costly and unstable score network inversion; $(ii)$ due to its optimization nature, it is less sensitive (and more interpretable) to hyperparameter tuning as opposed to DPS; $(iii)$ it is flexible about the observation noise that appears in the reconstruction loss $\mathbb{E}_{q(x_0|y)}[-\log p(y|x_0)]$. For instance it can easily handle Poisson noise as as well as Gaussian noise as discussed in section D.8. This in contrast with DPS that needs to approximate the likelihood $\log p(y|x_t)$, and thus resorts to approximation which require strict assumptions. For instance, for Poisson noise, it requires the signal intensity to be large enough so Poisson is approximated with Gaussian distribution. Last but not least, our variational framework allows forming more expressive distributions for the posterior $p(x_0|y)$ (e.g., hierarchical, multimodal, or normalizing flow based as done in Vahdat et al. (2021)). However, we leave this to future work.

