# OpenReview forum: "A Variational Perspective on Solving Inverse Problems with Diffusion Models"
_ICLR.cc/2024/Conference — ICLR 2024 poster_

### Official Review · Reviewer_5Auf · 2023-10-26

**Soundness:** 2 fair
**Presentation:** 2 fair
**Contribution:** 3 good
**Rating:** 5
**Confidence:** 4

**Summary:**

In this paper, the authors propose a novel diffusion sampling algorithm incorporating a data-fidelity term, allowing to solve inverse imaging problems with pretrained diffusion models. Interestingly, the authors claim that this novel sampling scheme can be related to a variational formulation, as opposed to traditional diffusion algorithms. This aspect is all the more interesting as it clearly relates the method to traditional, conventional variational methods for solving inverse problems. Eventually, the authors demonstrate its effectiveness on several inverse imaging problems, ranging from image inpainting to MRI imaging.

**Strengths:**

- The paper is overall well written and easy to follow, making it comfortable to read. The authors do not engage in unnecessary technicalities and provide the required background in a succint and clear manner.
- The proposed method is novel (to the best of my knowledge) and simple to implement.
- Relating the proposed method to the RED approach is very interesting and is a valuable contribution to the community.
- Experimental results are convincing.

**Weaknesses:**

Despite the many strong points raised above, this paper suffers from important drawbacks.

- Important references from the variational [1,2,3] and PnP [4,5,6] litterature are missing. Overall, while the context within diffusion models is well set, the link with the general imaging litterature is completely absent, making it difficult for the reader to relate to other imaging techniques.
- Theoretically, the fact that the Jacobian of the network is not symmetric (a priori) prevents the authors from making a link with a variational loss (see the Reehorst reference cited by the authors), questioning the full approach.
- Comparisons with DiffPIR [8] or RED are missing.

**References**

[8] Zhu Y, Zhang K, Liang J, Cao J, Wen B, Timofte R, Van Gool L. Denoising Diffusion Models for Plug-and-Play Image Restoration. In Proceedings of the IEEE/CVF Conference on Computer Vision and Pattern Recognition 2023

**Questions:**

I've enjoyed reading this paper and I thank the authors for their interesting work. I have however some strong feelings regarding how general imaging methods are treated in the paper - and in particular how their methods relates to more general variational methods. Please find below a list of comments.

**Major comments**
1. Pure variational methods are simply completely absent of this paper. Not only is this problematic with respect to the conveyed message ("the proposed method links with variational methods") but it is also detrimental to the experiments - for instance top row of Figure 11, a standard TV method would certainly reach very high PSNR. In the current form, I disagree that the paper links with variational methods; the paper links with RED approaches (which are very specific). Here are some more general references that may be of interest: [1,2,3].
2. The RED approach plays an important role in this paper, but the authors do not mention an important aspect of RED: the loss that it derives is incorrect. This is explicitely stated in the Reehorst and Schnitter reference cited by the authors. An important point raised there is that without symmetry of the Jacobian, there is no hope that the denoiser approximates a gradient. This is well known in the PnP community which has led to significant works [4,5,6].
3. Talking about fixed point after (9) is very misleading. In general, a fixed point is derived from the 0 of a subdifferential (or equivalently in the convex case, of the minimum of a convex function). In the case proposed by the authors, the loss varies with each iterate. How can there be a fixed point? (by the way, "the fixed point" strongly suggests a relationship to a convex function, switching to "a fixed point" would be more appropriate).
4. While the context on diffusion models is clear and the literature review is appropriate, this cannot be said on more general inverse imaging techniques. The authors do not seem to be aware of state-of-the-art PnP algorithms such as DPIR [7], and more importantly, to methods mixing PnP and diffusion techniques, such as DiffPIR [8]. I think comparing with at least one PnP would be interesting (e.g. DPIR, or since a strong emphasis is put on it, RED).
5. In my opinion, a key contribution of the paper is equation (8). An interesting point there is that the loss function is on $\mu$, which relates to Minimum Mean Square Error (MMSE) approaches in imaging. This is an important point as most traditional methods in imaging follow a Maximum a Posteriori (MAP) approach. In fact, it is often believed that MMSE approaches are better than MAP approaches, but unfortuntely too costly and/or difficult to implement. A comment on that would be very necessary and interesting to the imaging community readers.
6. The presence of $\epsilon$ in the equation of Proposition 2 is surprising given the remark made at the end of the proof in the appendix (just before A.3). Why keep it in the equation?
7. Maybe the authors could merge the results from A.2 and A.3 in Proposition 2 with additional notations regrouping the different constants. In fact, A.3 is an interesting result that is a bit hidden in the paper at the moment.

**Minor comments**
1. Small typos are remaining here and there. For instance: "stropped", "bellows up", "approaches to zero; ." (point after ;)...
2. "An inverse problem is often formulated as" --> "Inverse problems can be formulated as"

**References**

[1] Knoll F, Bredies K, Pock T, Stollberger R. Second order total generalized variation (TGV) for MRI. Magn Reson Med. 2011

[2] Portilla J, Strela V, Wainwright MJ, Simoncelli EP. Image denoising using scale mixtures of Gaussians in the wavelet domain. IEEE Transactions on Image processing. 2003

[3] Kobler E, Klatzer T, Hammernik K, Pock T. Variational networks: connecting variational methods and deep learning. InPattern Recognition: 39th German Conference, GCPR 2017, Basel, Switzerland, September 12–15, 2017.

[4] Hurault, Samuel, Arthur Leclaire, and Nicolas Papadakis. "Gradient Step Denoiser for convergent Plug-and-Play." International Conference on Learning Representations. 2021.

[5] Xu X, Sun Y, Liu J, Wohlberg B, Kamilov US. Provable convergence of plug-and-play priors with MMSE denoisers. IEEE Signal Processing Letters. 2020

[6] Pesquet JC, Repetti A, Terris M, Wiaux Y. Learning maximally monotone operators for image recovery. SIAM Journal on Imaging Sciences. 2021

[7] Zhang K, Li Y, Zuo W, Zhang L, Van Gool L, Timofte R. Plug-and-play image restoration with deep denoiser prior. IEEE Transactions on Pattern Analysis and Machine Intelligence. 2021

[8] Zhu Y, Zhang K, Liang J, Cao J, Wen B, Timofte R, Van Gool L. Denoising Diffusion Models for Plug-and-Play Image Restoration. In Proceedings of the IEEE/CVF Conference on Computer Vision and Pattern Recognition 2023

---

> ### Author Response · Authors · 2023-11-22
>
> We thank the reviewer for the positive opinion and constructive feedback especially about implications for imaging problems. Please find the comments addressed below:
>
> **Q1. missing references; while the context within diffusion models is well set, the link with the general imaging literature is completely absent.**
>
> **A1**. Thanks for pointing out the references that are all cited in the revised paper, see updated section C.1, and D.6.
>
> **Q2. the fact that the Jacobian of the network is not symmetric (a priori) prevents the authors from making a link with a variational loss, questioning the full approach.**
>
> **A2**. This is a good point. Please see our response to Q2 for the reviewer xtWM. Indeed, there is **NO** need for Jacobian symmetry. RED-diff only needs the gradient from Proposition 2. We used the explicit regularization term only to form a simple and intuitive objective for automatic differentiation, which can also work directly with gradient. We clarified this in section 4.2 the revised paper.
>
> **Q3. Comparisons with DiffPIR [8] or RED are missing.**
>
> **A3**. We have added a new section D.6 to the revised paper to compare with PnP methods: DPIR and DiffPIR. Our method outperforms DPIR and achieves slightly better performance than DiffPIR. Note that RED-Diff is derived from a maximum likelihood perspective and the MAP estimation, and it provides a rigorous treatment for the problem. For RED there is no public codebase that we can run with their setting. We have comparisons included with a single-denoiser version of our work that is closer to RED; see section D.5.
>
>
> **Q4. Pure variational methods absent. Not only is this problematic with respect to the conveyed message but it is also detrimental to the experiments. I disagree that the paper links with variational methods; the paper links with RED approaches; more general references that may be of interest: [1,2,3].**
>
> **A4**: We present the paper for the no dispersion case ($\sigma=0$) for simplicity of exposition. We agree that $\sigma=0$, leads to RED-type methods. However, an extension to the variational Gaussian (with non-zero sigma) is already presented in section A. 3 of the appendix. We also believe in general, with the current framework, one can consider more complex (multimodal) distributions modeled e.g., via hierarchical or normalizing flow-based models as in VAEs, following the procedure in previous work; see e.g., [G]. We clarified this point in the revised version; see new section D.9. Thanks also for pointing out the references that are cited in the revised version.
>
> **Q5. incorrect loss for RED;  stated in the Reehorst and Schnitter reference. An important point raised there is that without symmetry of the Jacobian, there is no hope that the denoiser approximates a gradient. This is well known in the PnP community which has led to significant works [4,5,6].**
>
> **A5**. We agree with the reviewer. As stated in response to Q2, we **DO NOT** need any explicit regularization and thus no Jacobian symmetry for the RED-diff algorithm to work. RED-diff only needs the gradient in Proposition 2.
>
> **Q6. Talking about fixed point after (9) misleading because the loss varies with each iterate. How can there be a fixed point?**
>
> **A6**. We use the term 'fixed point' in a figurative manner. We need a fixed point interpretation in the expectation sense. This needs a careful examination and is left for furniture work. We rephrase it to ensure clarity and avoid potential misunderstandings.
>
> **Q7. insufficient context about inverse imaging techniques.; not aware of state-of-the-art PnP algorithms such as DPIR [7] and DiffPIR [8]. I think comparing with at least one PnP would be interesting (e.g. DPIR, or since a strong emphasis is put on it, RED).**
>
> **A7**. Thanks for pointing out the references from PnP literature that are cited now. We have added a new section D. 6 in the revised paper to compare with PnP methods including DPIR and DiffPIR. For inpainting task, the gains of RED-diff are obvious compared with DPIR. The performance is slightly better than DiffPIR; see table im response to Q4 for reviewer xtWM.
>
> **Q8. The presence of $\epsilon$ in Proposition 2 is surprising given the remark made at the end of the proof in the appendix. Why keep the equation?**
>
> **A8**. it can be removed. We keep it because $\epsilon_{\theta} - \epsilon$ resembles the gradient (difference between denoisers) in RED. It’s also insightful for variance reduction methods to choose alternatives for $\epsilon$ that can reduce variance for better convergence.
>
> **Q9.merge the results from A.2 and A.3 in Proposition 2 with additional notations regrouping the different constants. A.3 is an interesting result that is a bit hidden.**
>
> **A9**. We rather keep the main paper simple by presenting the simple scenario with sigma=0. We add statements in the main paper to discuss A.3.
>
> [G] Vahdat et al. “Score-based generative modeling in latent space.” In NeurIPS, 2021.

---

### Official Review · Reviewer_RLDf · 2023-10-29

**Soundness:** 2 fair
**Presentation:** 3 good
**Contribution:** 3 good
**Rating:** 5
**Confidence:** 4

**Summary:**

In this paper, the authors propose a new regularization strategy for solving inverse problems with a denoising prior based on diffusion models. The regularization is reminiscent to RED but uses denoisers at different noise levels. Compared to competitive methods, DiffRED does not assume the posterior $p(x_0|x_t)$ of the diffusion model to be unimodal. Experiments on various inverse problems prove the efficiency of the algorithm.

**Strengths:**

Overall, the paper is well written, the motivation is clear and the method is original. The main strength of the paper is its experimental study. The authors experimented on very diverse inverse problems and perform an exhaustive ablation study. They clearly demonstrate that their algorithm performs state-of-the art performance and gives impressive visual results. Concerning the method, the proposed regularization in (8) is new and original, and the Proposition 2 makes this regularization term very promising.

**Weaknesses:**

My main concerns are on the theoretical side. The following aspects require further details.

- You introduce KL minimization in (5) which makes sense for sampling from $p(x_0|y)$. However, the Gaussian approximating posterior $q(x_0|y)$ is then used with $\sigma = 0$. In this case, the minimization of the KL comes back to the MAP problem $argmin_\mu p(\mu|y) = argmin_\mu p(y|\mu) - \log p(\mu) $. Therefore, the objective of this work is then to solve the MAP. This is a classical link between sampling and optimization.  From this observation, assuming Proposition 1 is true, by identification in the case $\sigma = 0$, we get that $-\log p(\mu)$ is equal to the second term in (8). Is this really true ? In general, can the authors explain why they first consider a sampling approach before taking $\sigma = 0$ ?
- The proof of Proposition 1 uses a result from (Song et. al , 2021) that seems different (inequality and not equality). As this is the base of the theoretical analysis, I would like to see an exhaustive proof of the result.
- The proposed regularization is defined with an expectation on $t$, but in practice $t$ is not chosen at random but with a predefined descending scheme. Therefore, the algorithm and the regularization used in practice is different from the ones originally introduced. This is for me the main weakness of the paper.
- Proposition 2 implies that the right term in the equation is a conservative vector field, and in particular that it has symmetric Jacobian. This looks surprising to me without further assumptions on $\epsilon_\theta$. Refer to (Reehorst & Shniter, 2018) for the same observation on RED. Do you have an argument for justifying symmetry of the Jacobian ?
- The algorithm 1 is proposed without any convergence analysis.
- Before (8) it should be made clear that you assume exact approximation of the score with $\epsilon_\theta$ to get (8).

**Questions:**

- The authors argue that the advantage of the paper is to avoid the assumption $p(x_0|x_t)$ unimodal. Instead, they take the assumption
$q(x_0|y)$ unimodal. Can the authors explain why this assumption would be more valid ?

---

> ### Author Response · Authors · 2023-11-22
>
> We thank the reviewer for the positive opinion and constructive feedback about the theoretical aspects. Please find the comments addressed below:
>
> **Q1. KL minimization in (5) makes sense for sampling from  $p(x_0|y)$. However, the Gaussian approximating posterior $q(x_0|y)$ is then used with $\sigma=0$. In this case, the minimization of the KL comes back to MAP. This is a classical link between sampling and optimization. From this observation, assuming Proposition 1 is true, by identification in the case $\sigma=0$, we get that $−\log⁡p(\mu)$ is equal to the second term in (8). Is this really true? can the authors explain why they first consider a sampling approach before taking  $\sigma=0$?**
>
> **A1.** This is correct. If $\sigma=0$, KL minimization boils down to MAP estimation. We present this scenario for simplicity in the main paper, but the extension to the nonzero dispersion ($\sigma$ > 0) is already presented in section A.3 of the appendix. Having said that, we believe our framework can be extended to more complex variational distributions such as hierarchical, multimodal, or normalizing flow-based distributions as in VAEs, following the recipe in the previous work such as [G]. We clarify this in section A. 3 of the revised paper.
>
> **Q2: The proof of Proposition 1 uses a result from (Song et. al , 2021) that seems different (inequality and not equality). As this is the base of the theoretical analysis, I would like to see an exhaustive proof of the result.**
>
> **A2.** This is a great point. We assume the score function is exactly learned in the pre-training phase, namely $s_{\theta}(x)=\nabla \log p(x)$, where $p(x)$ is the true data distribution. This is an important assumption that’s used in Theorem 2 by Song et. al 2021. We will explicitly state this assumption in Proposition 1. We add clarifications to the proof section too.
>
> **Q3. The proposed regularization is defined with an expectation on $t$,, but in practice $t$ is not chosen at random but with a predefined descending scheme. Therefore, the algorithm and the regularization used in practice is different from the ones originally introduced.**
>
> **A3**: We agree that random sampling is a natural choice as we propose stochastic optimization solvers. The results for random sampling are presented in Fig. 16 (4th column). Designing a proper random sampling is an orthogonal angle and we leave it for future research.
>
> **Q4. Proposition 2 implies that the right term in the equation is a conservative vector field, and in particular that it has symmetric Jacobian. This looks surprising to me without further assumptions on $\epsilon_{\theta}$. Refer to (Reehorst & Shniter, 2018) for the same observation on RED. Do you have an argument for justifying symmetry of the Jacobian?**
>
> **A4**. It's not clear to us that r.h.s of Proposition 2 is a conservative vector field. Can you please clarify? We should note that we DO NOT need any explicit form for the regularization term, and only work with gradients in proposition 2, so no need for Jacobian symmetry (see response to Q2 for reviewer xtWM
> ).
>
> **Q5. The algorithm 1 is proposed without any convergence analysis. Before (8) it should be made clear that you assume an exact approximation of the score with $\epsilon_{\theta}$ to get (8).**
>
> **A5**. Yes. In the revised version, we explicitly state before Proposition 2 that $\epsilon_{\theta}$ is an exact approximation. Thanks for bringing up this important yet tacit point. Convergence analysis is beyond the scope of the current paper given the highly nonlinear and nonconvex optimization landscape.
>
> **Q6. unimodal assumption for $q(x_0|y)$. Why is this assumption more valid?**
>
> **A6**. We agree that variational inference is also a unimodal approximation. It however approximates the posterior $p(x_0|y)$, while previous work approximated the likelihood $p(x_0|x_t)$ that was certainly invalid for larger noise levels. The advantages of our posterior approximation are twofold: (1) it requires no score jacobian as opposed to DPS and PGDM that need costly and unstable score network inversion; (2) it is less sensitive (and more interpretable) to hyperparameter tuning as opposed to DPS. in addition, with posterior approximation one can relax unimodal assumption by using more expressive variational distributions. Please see new section D.9 for more elaboration.
>
> [G] Vahdat et al. “Score-based generative modeling in latent space.” In NeurIPS, 2021.

---

> ### Comment · Reviewer_RLDf · 2023-12-04
>
> Thanks for the detailed answer. However, some important doubts remain unanswered.
>
> There is sometimes an orthogonality between what is presented and what is done in practice:
> 1) Even if it is possible to extend the theory to $\sigma>0$, the authors only use the case  $\sigma=0$ in practice. Therefore, I am questioning the presentation of the work as a posterior sampling approach when it is actually solving the MAP.
> 2) The proposed regularization is defined with an expectation on the time $t$, but in practice $t$ is not chosen at random but with a well chosen scheme.
>
> Moreover:
> - Q2 : The proof has not been clarified by the authors. It is still not clear where (12) comes from. As this is the basis of the analysis, this is a strong issue.
> - The comparison with DPIR added during the rebuttal period seems unfair. It has been realized for inpainting large holes + with IRCNN denoiser. First, the use of the IRCNN denoiser does not make sense has the DPIR original paper is with the larger and more powerful DRUNet denoiser. The use of the IRCNN denoiser looks like to make DPIR underperform. Moreover, DPIR is likely to be sub-optimal when inpainting large holes, but should perform good on deblurring experiments (as experimented in DPIR original paper).
>
> I decide to keep my score.

---

### Official Review · Reviewer_Qren · 2023-10-30

**Soundness:** 3 good
**Presentation:** 3 good
**Contribution:** 3 good
**Rating:** 6
**Confidence:** 3

**Summary:**

This paper presents a sampling process based on diffusion models for solving inverse problems. This approach allows to tackle very general inverse problems in images like inpainting, deblurring, super-resolution... The method itself is straightforward when using a pretrained network and the experiments show promising results.

**Strengths:**

The method takes the form of a sampling algorithm that can solve a large number of inverse problems with additive Gaussian noise.
The framework is easy to setup when using pre-trained diffusion models and the theory gives access to formula for the hyperparameters. All in all the experiments show the benefit of this approach compare to similar frameworks.

**Weaknesses:**

The method is only for Gaussian noise while DPS, for example, can handle Poisson noise (though not in a low regime). Also the hyperparameter formula asks for the variance of the noise. Such quantity may be unknown, it would been interesting to see, experimentally, what happens with wrong or estimated values of the noise variance. Finally, like all methods based on diffusion models, it asks for an appropriate model (pre-trained or not) and solving the inverse problems remains computationally heavy with thousand of iterations to produce a result.

**Questions:**

I reformulate one of my previous remarks as a question: what happens with wrong or estimated values of the noise variance?

My next question is also a remark. The authors compare different strategy for the sampling (i.e. the timestep). The conclusion is that the descending sampling leads to the best results. This is normal as there is a time dependency in the sampler since the previous mu is used for the loss. Thus, I would expect the random sampling to fail. So the discussion on this point is weak and I would expect other descending strategy (log, exp...). Would it be a way to reduce the number of timesteps?

Last point, please update the references. For example most of the paper from Chung et al are proceedings of conferences and no just papers on ArXiv.

---

> ### Author Response · Authors · 2023-11-22
>
> We thank the reviewer for the positive comments and constructive feedback. Please find the comments addressed below:
>
> **Q1. The method is only for Gaussian noise while DPS can handle Poisson noise (though not in a low regime).**
>
> **A1.** We presented Gaussian observation noise only to simplify the exposition. In general, our framework is flexible with the choice of the observation noise distribution. The reconstruction term involves $\log p(y|x_0)$ as in eqn. (6). For the observation model $y \sim {\rm Poisson}(\lambda=f(x0))$, the likelihood obeys $p(y|x_0) = {\rm Poisson}(y; f(x_0))$ where $f(x_0)$ is the mean parameter. , Although $y$ can be discrete in the Poisson distribution, we find that $\log p(y|x_0)$ is a simple differentiable function that can be optimized via automatic differentiation libraries (e.g., PyTorch). Note that this is yet another advantage of RED-diff compared with DPS, as DPS needs to find $\log p(y|x_t)$ per diffusion timestep and uses aGaussian approximation of Poisson distribution, which holds only in the high intensity regime. We added a new section D.7 in the appendix to elaborate on this.
>
> **Q2. I reformulate one of my previous remarks as a question: what happens with wrong or estimated values of the noise variance?**
>
> **A2**. The observation noise variance is absorbed to the regularization weight $\lambda$. We tune $\lambda$ based on “validation data” for the best performance. Please see Fig. 15(left) which plots KID metric versus lambda.
>
> **Q3. Finally, like all methods based on diffusion models, it asks for an appropriate model (pre-trained or not) and solving the inverse problems remains computationally heavy with thousands of iterations to produce a result.**
>
> **A3**. We agree that RED-diff is iterative by nature. However, our experiments show the best performance achieved with 100 iterations or less. RED-diff can be accelerated via distillation methods, amortization techniques, or better optimizers. It is however an orthogonal angle, beyond the scope of the current submission.
>
> **Q4: The authors compare different strategies for the sampling (i.e. the timestep). The conclusion is that the descending sampling leads to the best results. This is normal as there is a time dependency in the sampler since the previous mu is used for the loss. Thus, I would expect the random sampling to fail. So the discussion on this point is weak and I would expect other descending strategy (log, exp...). Would it be a way to reduce the number of timesteps?**
>
> **A4**: Since our algorithm is derived based on stochastic optimization for expectation minimization, random sampling is a natural choice to arrive at unbiased estimates. Exploring timestepping for descending strategy is a good point. As per reviewer’s suggestion, we compare linear/log/exp timestepping schedules, and the results are reported in the new section D.8 of the revised paper. In conclusion, the linear schedule seems to be the best strategy.
>
> **Q5: Last point, please update the references. For example most of the paper from Chung et al are proceedings of conferences and not just papers on ArXiv.**
>
> **A5**: Thanks for pointing this out. References are updated in the revised paper.

---

### Official Review · Reviewer_xtWM · 2023-10-30

**Soundness:** 3 good
**Presentation:** 3 good
**Contribution:** 3 good
**Rating:** 6
**Confidence:** 5

**Summary:**

This paper presents a novel stochastic adaptation of regularization by denoising (RED) using the denoising diffusion model as an image prior. The study demonstrates the derivation of the score matching loss under the variational inference framework, which naturally leads to the formulation of stochastic RED. By implementing an appropriate weighting scheme, the proposed variational sampler surpasses existing diffusion-based posterior samplers in numerous inverse imaging problems, exhibiting superior reconstruction faithfulness (measured by PSNR/SSIM) and enhanced visual quality (evaluated via KID/LPIPS).

**Strengths:**

The paper is clearly written and comprehensible to readers. Additionally, the formulation of the proposed variational sampler, utilizing a first-order stochastic optimizer (Adam) to address the stochastic RED objective, is both innovative and straightforward, yielding effective results.

**Weaknesses:**

Despite the paper's clarity, several imprecise arguments and overstatements necessitate revision and clarification:

- The authors claim that a key advantage of their method is the circumvention of unimodal estimation employed in prior works. However, their variational inference approach introduces another level of approximation by using a simple unimodal Gaussian to approximate the complex multi-modal conditional posterior p(x0​∣y). This complicates the justification of the proposed framework's improvement over mitigating posterior score approximation in prior methods (Contribution 1).
- The claim that the RED framework fundamentally differs from the Plug-and-Play (PnP) framework lacks rigor. The explicit regularization term in RED only exists under certain strict assumptions, which are often not applicable to modern deep learning-based denoisers. Thus, both RED and PnP should be explained from a fixed-point iteration perspective, which provides a unified treatment with theoretical convergence guarantees [A].
- The proposed weighting mechanism aligns with the parameter setting strategy in the PnP framework [B] (as detailed in Section 4.2). Moreover, the authors overlook discussing several related works in PnP-type literature [C, D, E], which essentially represent the deterministic counterparts of diffusion-based posterior sampling.
- The experimental comparison with DPS lacks sufficiency to verify the effectiveness of the proposed approach. Benchmarking against the state-of-the-art PnP approach [B] would provide a more comprehensive characterization of the method's advantages. Additionally, the performance in phase retrieval, particularly the PSNR, is too weak to be considered meaningful. To enhance credibility, it is recommended to consider adopting more realistic settings, such as coded diffraction patterns or oversampled Fourier measurements, as in [F].

**References:**

[A] Regev Cohen, Michael Elad, and Peyman Milanfar. "Regularization by denoising via fixed-point projection (red-pro)." SIAM Journal on Imaging Sciences, 14(3):1374–1406, 2021.
[B] Zhang, Kai, et al. "Plug-and-play image restoration with deep denoiser prior." IEEE Transactions on Pattern Analysis and Machine Intelligence 44.10 (2021): 6360-6376.
[C] Kamilov, Ulugbek S., et al. "Plug-and-play methods for integrating physical and learned models in computational imaging: Theory, algorithms, and applications." IEEE Signal Processing Magazine 40.1 (2023): 85-97.
[D] Wei, Kaixuan, et al. "TFPNP: Tuning-free plug-and-play proximal algorithms with applications to inverse imaging problems." The Journal of Machine Learning Research 23.1 (2022): 699-746.
[E] Laumont, Rémi, et al. "Bayesian imaging using plug & play priors: when langevin meets tweedie." SIAM Journal on Imaging Sciences 15.2 (2022): 701-737.
[F] Metzler, Christopher, et al. "prDeep: Robust phase retrieval with a flexible deep network." International Conference on Machine Learning. PMLR, 2018.

**Questions:**

See above

---
After rebuttal: I'm generally satisfied with authors' response, overall I think it's a good paper that can be accepted to ICLR.

---

> ### Author Response · Authors · 2023-11-22
>
> We thank the reviewer for the positive comments and the constructive feedback. Please find the comments addressed below. Thanks also for pointing out the missing references A-F that are cited now.
>
> **Q1: Unimodal approximation of the posterior, improvement over prior methods?**
>
> **A1**: We totally agree that our variational approximation is a unimodal approximation of the posterior. However, the advantages of new approximation are twofold: (1) it needs no score jacobian as opposed to DPS and PGDM that need costly and unstable score network inversion; (2) it is less sensitive and more interpretable for hyperparameter tuning as opposed to DPS. This is validated by extensive experiments reported in the paper. Last but not least, our variational framework allows forming more expressive distributions for the posterior $p(x_0​∣y)$ (e.g., hierarchical, multimodal, or normalizing flow based as done in [G]). However, we leave this to future work. We elaborate this in a new section D. 9 of the revised paper. The corresponding statement in section 3 is also rephrased.
>
> **Q2: lack of rigor for comparison of RED and PnP framework; explicit regularization term in RED only exists under certain strict assumptions;  both RED and PnP should be explained from a fixed-point iteration perspective [A].**
>
> **A2:** Sorry for the confusion. We should note that we **DO NOT** need any explicit regularization for RED-diff to work. At the end of the day, one only needs the gradient term (as in Proposition 2) to run RED-diff. We simply presented the explicit regularization term $<{\rm sg}(\epsilon_{\theta}(x_t))]-\epsilon), \mu >$ in Algorithm 1 to form an optimization objective for automatic differentiation-based training. Note that there is a ${\rm sg}$ (stopped-gradient), which means the gradient is always $\epsilon_{\theta}(x_t) - \epsilon$, with no need for Jacobian symmetry of the denoiser. As a result, based on \[H\], no strict assumptions are required. We further clarify this in section 4.2 of the revised paper.
>
> We agree that RED and PnP frameworks are inter-related via fixed-point iterations as in [A]. Indeed, since RED-diff only works with the gradients as in Proposition 2, it can be treated as a fixed point solution. However, RED-diff is generative (stochastic) and thus fixed-point interpretation needs to be established in the expectation sense. This needs more careful examination and we leave it for future work.
>
> To clarify this, we rephrase the statement about the difference between RED and PnP framework in section 2.
>
> **Q3: proposed weighting mechanism aligns with the parameter setting strategy in [B]; overlook discussing several related works in PnP-type literature [C, D, E], which represent deterministic counterparts of diffusion-based sampling.**
>
> **A3**: Thanks for pointing out the references A-F that are now cited in the revised version. Our weighting mechanism calibrateS denosier of diffusion models, and it is derived based on MMSE estimation and Tweetie’s formula, which is in a completely different context than [B].
>
> **Q4: experimental comparison with DPS lacks sufficiency;  benchmarking against PnP [B]; performance in phase retrieval, PSNR too weak.**
>
> **A4**: We add a new section in the revised manuscript and compare RED-diff with PnP methods including DPIR [B] (and its generative version DiffPIR [K]) for inpainting ImageNet data. It’s seen that RED-diff significantly outperforms DPIR due to its generative diffusion prior. RED-diff is also better and to some extent on par with DiffPIR. We report quantitative results here and elaborated in section D. 6 of appendix.
>
> Table 7: Comparison with PnP methods for image inpainting for Imagenet data.
>
> | Sampler   | PSNR(dB) $\uparrow$ | SSIM $\uparrow$ | KID $\downarrow$ | LPIPS $\downarrow$ | top-1 $\uparrow$ |
> |-----------|----------------------|-----------------|-------------------|--------------------|-------------------|
> | DPIR      | 16.50                | 0.792           | 42.62             | 0.237              | 60.0              |
> | DiffPIR   | **27.29**            | 0.8728          | 7.96              | 0.1401             | 70.0              |
> | RED-diff  | 26.06                | **0.8792**      | **7.92**          | **0.1060**          | **90.0**          |
>
> Regarding the phase retrieval problem, you are making a great point. We need to limit the measurement model to specific measurements (e.g., Gaussian or coded diffraction patterns) to achieve better quality and make a better assessment. However, that requires its own study beyond the scope of this submission. We add clarifying statements in section 5.2.
>
> [G] Vahdat et al. “Score-based generative modeling in latent space.” NeurIPS, 2021.
>
> [H] Reehorst, E.T. and Schniter, P., 2018. Regularization by denoising: Clarifications and new interpretations. IEEE trans on computational imaging.
>
> [K] Zhu Y et al. Denoising Diffusion Models for Plug-and-Play Image Restoration. CVPR workshop 2023.

---

### Meta-Review · Area_Chair_XyHt · 2023-12-08

**Metareview:**

This paper introduces a method for solving inverse imaging problems using diffusion models. The key innovation is the integration of a denoising diffusion process as a prior in a variational framework. This method, termed RED-diff, uses denoisers at various timesteps to impose distinct structural constraints on images. A signal-to-noise ratio (SNR) based weighting mechanism is proposed to optimize the contribution of denoisers from different timesteps. Unlike more standard methods the sampling is formulated as a stochastic optimization procedure. The experiments (restauration / inpainting deblurring / super-resolution) seem to indicate that the proposed method outperforms existing diffusion-based methods under several metrics.

**Justification For Why Not Higher Score:**

The manuscript presents novel concepts, as recognized by most reviewers. However, the overall methodology is somewhat incremental and, as such, does not justify a spotlight presentation.

**Justification For Why Not Lower Score:**

The manuscript presents novel concepts, as recognized by most reviewers, and is mostly clearly written and well-motivated. The numerical experiments are solid and do show that the methodology, in some specific contexts, improves upon the SoTA.

---

### Decision · Program_Chairs · 2024-01-16

Accept (poster)